# What Data Enables Optimal Decisions?
# An Exact Characterization for Linear Optimization

**Omar Bennouna**
MIT
omarben@mit.edu

**Amine Bennouna**
Northwestern University
amine.bennouna@northwestern.edu

**Saurabh Amin**
MIT
amins@mit.edu

**Asuman Ozdaglar**
MIT
asuman@mit.edu

## Abstract

We study the fundamental question of how informative a dataset is for solving a given decision-making task. In our setting, the dataset provides partial information about unknown parameters that influence task outcomes. Focusing on linear programs, we characterize when a dataset is sufficient to recover an optimal decision, given an uncertainty set on the cost vector. Our main contribution is a sharp geometric characterization that identifies the directions of the cost vector that matter for optimality, relative to the task constraints and uncertainty set. We further develop a practical algorithm that, for a given task, constructs a minimal or least-costly sufficient dataset. Our results reveal that small, well-chosen datasets can often fully determine optimal decisions—offering a principled foundation for task-aware data selection.

## 1 Introduction

Decision-making problems are often performed under incomplete knowledge of the state of nature—that is, they rely on parameters that must be learned or estimated. In practice, experts draw on a combination of domain knowledge and experience from previously solved tasks. With the recent surge in data availability, data-driven decision-making has become a dominant paradigm: data now plays a central role in complementing contextual knowledge to guide decisions. This paper seeks to understand the informational value of a given dataset with respect to a specific decision-making task. More precisely, we ask: to what extent does a dataset enable recovery of the optimal decision, given task structure and prior knowledge?

The fundamental question of data informativeness—or its value—has several important implications. One key implication is data collection: when faced with a new decision-making task, which data should be collected to effectively generalize prior knowledge to the new setting? Ideally, one seeks the smallest—or least costly—yet most informative dataset. A second major implication lies in computing. Recent successes of large-scale models (e.g., LLMs) have been driven by large-scale data and advances in computing. However, computing cost remains a significant bottleneck. Identifying the most informative subset of data for a specific task can significantly reduce dataset size and, consequently, training costs. Quantifying data value also impacts mechanism design in data markets and considerations around privacy.

A general setting for studying this question is as follows. Suppose the decision-maker's goal is to select a decision $x \in \mathcal{X}$ minimizing a loss $L(x, \theta)$ which depends on an unknown parameter—state of nature—$\theta \in \Theta$. A dataset $\mathcal{D} = \{q_1, \dots, q_N\}$ consists of points at which the loss is evaluated: it

39th Conference on Neural Information Processing Systems (NeurIPS 2025).

provides observations $\{L(q_1, \theta) + \epsilon(q_1, \theta), \ldots, L(q_N, \theta) + \epsilon(q_N, \theta)\}$ partially informing on the true state of nature $\theta$, where the random variable $\epsilon(\cdot, \cdot)$ models noisy observations. The decision-maker can then use the observations along with their prior knowledge (set restriction $\theta \in \Theta$) to select a decision $x \in \mathcal{X}$ minimizing $L(x, \theta)$. The central question is: which datasets $\mathcal{D}$ allow to recover the task-optimal decision, given the prior knowledge encoded in the uncertainty set $\Theta$. In the rest of the paper, we study this question in the setting of linear programming—with linear loss $L$ and polyhedral decision set $\mathcal{X}$—where the task structure enables a sharp analysis.

To illustrate this formalism with an example, consider a hiring problem in which a decision-maker is given a list of candidates and their resumes and decides which subset of candidates to interview in order to reveal their value, and ultimately make a hiring decision. This problem has been studied in various settings (Purohit et al. 2019, Epstein and Ma 2024), including within the popular Secretary Problem (Kleinberg 2005, Arlotto and Gurvich 2019, Bray 2019). Prior work typically assumes a sequential, adaptive model, where interviews and hiring decisions occur in an online fashion. However, in many real-world scenarios—such as hiring PhD students or faculty—the set of candidates to interview must be chosen in advance, with hiring decisions made afterward based on all interview outcomes. This latter *offline* setting is a natural instance of the data informativeness problem.

Formally, hiring from $d$ candidates is a decision-making problem where a decision consists of a binary vector $x \in \mathcal{X} \subset \{0, 1\}^d$ indicating which candidates to hire. The feasible set $\mathcal{X}$ encodes organizational constraints, such as a maximum number of hires $\sum_{i=1}^{d} x_i \leq k$, or maximum expertise-based quotas $\sum_{i \in I_j} x_i \leq k_j$ for subsets $I_j \subseteq [d]$, to name a few. Each candidate has an unknown value $\theta_i$, with $\theta \in \Theta \subset \mathbb{R}^d$ modeling prior information on these values. It consists here in (i) candidates' resumes, which can be seen as features $\phi = (\phi_1, \ldots, \phi_d) \in \mathbb{R}^{l \times d}$, and (ii) historical hiring data, which is pairs of resumes and observed value $(\hat{\phi}_1, \hat{\theta}_1), \ldots, (\hat{\phi}_l, \hat{\theta}_n)$. Specifically, $\Theta = \{\theta \in \mathbb{R}_+^d \ : \ \exists \alpha \in \mathbb{R}^l, \exists \epsilon, \epsilon' \in \mathcal{E}, \mathcal{E}' \text{ s.t. } \theta = \alpha^\top \phi + \epsilon, \ \hat{\theta} = \alpha^\top \hat{\phi} + \epsilon'\}$ with $\mathcal{E} \subset \mathbb{R}^d, \mathcal{E}' \subset \mathbb{R}^l$ noise sets. The loss incurred by a decision $x$ under values $\theta$ is $L(x, \theta) = -\theta^\top x$—the negative total value of selected candidates. A dataset $\mathcal{D} \subset \{q \in \{0, 1\}^d \ : \ \sum_{i=1}^{d} q_i = 1\}$ is a subset of candidates to interview, and each interview $q \in \mathcal{D}, q_j = 1$, reveals a, possibly noisy, evaluation of a given candidate $j$'s value $L(q, \theta) = \theta_j$ which complements the prior information embedded in $\Theta$. The goal in this application is to select the smallest subset of candidates to interview (dataset) to recover the optimal hiring decision: that is, the smallest, informative dataset for the given task.

The question of data informativeness is related to several extensively studied topics in economics, statistics, computer science, and operations research literature. Below, we highlight a few of these areas and the angle with which they approached this question.

**Active Learning, Bandits and Adaptive Experimental Design.** In many data-driven settings, informativeness is approached via adaptive, sequential data collection. Active learning (Settles 2009, Zheng et al. 2017) seeks to sequentially select data points that improve a classifier by minimizing predictive loss, while bandit algorithms (Lattimore and Szepesvári 2020, Carlsson et al. 2024) aim to optimize decisions through sequential exploration. Adaptive experimental design (Zhao 2024) similarly selects experiments to maximize information gain about unknown parameters, often guided by Bayesian criteria such as posterior variance reduction. These approaches rely on real-time feedback to guide data acquisition and typically analyze asymptotic behavior. However, in practical applications—such as surveys or field trials—queries must often be selected in advance, and outcomes are revealed only afterward. In such settings, adaptivity is infeasible.

In contrast to these paradigms, we study fixed datasets in a non-adaptive, finite-sample regime, focusing on geometric conditions for optimal decision recovery rather than statistical estimation error. We show that, even without adaptivity, one can precisely characterize which datasets are sufficient to recover task-optimal decisions—offering an offline analogue to adaptive data selection.

**Blackwell's Informativeness Theory.** One of the earliest and most celebrated frameworks for comparing datasets is Blackwell's theory of informativeness (Blackwell 1953). In this framework, a dataset is abstracted as an experiment, which generates a signal $s \in S$ drawn from a distribution $P(s|\theta)$, informing on $\theta \in \Theta$, the unknown state of nature. This is equivalent to our framing above, with the signal being $s = (L(q_1, \theta) + \epsilon(q_1, \theta), \ldots, L(q_N, \theta) + \epsilon(q_N, \theta))$ and the noise terms specifying the conditional distribution. Two datasets (experiments) are compared by whether one enables better decision-making across *all loss functions* and priors. Formally, an experiment $P$ is

more informative than experiment $Q$ if

$$\inf_{\delta:S\to\mathcal{X}} \mathbb{E}_{\theta\sim\pi,\, s\sim P(\cdot|\theta)}[L(\delta(s),\theta)] \leq \inf_{\delta:S\to\mathcal{X}} \mathbb{E}_{\theta\sim\pi,\, s\sim Q(\cdot|\theta)}[L(\delta(s),\theta)], \quad \text{for all loss } L \text{ and prior } \pi$$

Blackwell's seminal result shows that this criterion is equivalent to several elegant characterizations, notably through the notion of garbling (de Oliveira 2018).

Blackwell's informativeness criterion imposes a strict requirement: it compares datasets by whether they enable better decisions across *all possible tasks*. In contrast, our work fixes the decision task (loss function $L$ and structure $\mathcal{X}$) and asks which datasets suffice to recover the task-optimal decision. This restriction aligns better with practical applications but also makes the informativeness question more delicate: as Le Cam (1996) observed, such questions may become "complex or impossible depending on the statistician's goal". Whereas Blackwell compares datasets by their universal utility, our work develops a tractable, task-specific notion of informativeness grounded in the structure of the decision task itself.

**Influence Functions and Robust Statistics.** Influence functions, originating in robust statistics (Huber 1992, Hampel et al. 1986), quantify the local impact of individual data points on estimators and have recently received renewed interest (Broderick et al. 2023). Similar approaches include DataShapely (Ghorbani and Zou 2019, Kwon and Zou 2022, Jiang et al. 2023, Jia et al. 2023), and Datamodels (Ilyas et al. 2022, Dass et al. 2025, Ilyas and Engstrom 2025). These methods typically analyze how *small perturbations* to a dataset affect the output of a *fixed estimator*. However, a key limitation of this approach is that data value is generally "non-additive": the informativeness of an individual data point is not intrinsic, but rather related to the data set as a whole. Our focus is on the joint informativeness of the full dataset—characterizing when a collection of observations, as a whole, suffices to recover the task-optimal decision. Joint informativeness, combinatorial in nature, is a more challenging problem (Rubinstein and Hopkins 2024, Freund and Hopkins 2023). Furthermore, while influence functions assess sensitivity in *estimation* problems under *fixed* inference procedures, our framework evaluates data informativeness with respect to a *decision task*, at a dataset-level *independently* of any specific inference or optimization procedure.

Data informativeness is a fundamental problem that relates to multiple literature streams—such as Stochastic Probing (Weitzman 1979, Singla 2018, Gallego and Segev 2022), Optimal Experimental Design (Chaloner and Verdinelli 1995, Singh and Xie 2020) and Algorithms with Predictions (Mitzenmacher and Vassilvitskii 2020)—but a detailed comparison is beyond the scope of this paper.

**Contributions.** This paper addresses the problem of evaluating the informativeness of datasets relative to a specific decision-making task. We study informativeness in the sense of being able to recover the task's optimal solution. This problem is challenging: it is combinatorial in nature, requiring assessment of the value of different combinations of data points. Moreover, informativeness in decision-making is difficult to quantify. One must identify how information in a dataset is relevant to decisions in the feasible set $\mathcal{X}$, relative to prior information encoded in the uncertainty set $\Theta$.

To be able to derive precise insights, we focus on tasks that can be formulated as linear programs—a broad and expressive class of decision-making problems whose geometric structure enables precise theoretical analysis. Our main contributions are as follows:

- **Geometric Characterization of Dataset Sufficiency:** We prove a necessary and sufficient condition (Theorem 1) under which a dataset is *sufficient* to recover the optimal decision for a linear program under cost uncertainty. This condition is framed geometrically: a dataset is sufficient if it spans the task-relevant directions that govern what can change the optimal solution, given the structure of the feasible set $\mathcal{X}$ and the uncertainty set $\Theta$.

- **Constructive Characterization via Reachable Optimal Solutions:** We show that the span of relevant directions for dataset sufficiency can equivalently be expressed as the span of differences between optimal solutions under different cost vectors in the uncertainty set. This characterization (Theorem 2) provides an algorithmically accessible formulation for evaluating and constructing sufficient datasets.

- **Efficient Data Collection Algorithm:** Building on these characterizations, we develop an iterative algorithm that constructs a minimal sufficient dataset. When the uncertainty set is polyhedral, the algorithm terminates in a number of steps equal to the size of the minimal sufficient dataset, and each step involves solving a tractable mixed-integer program.

## 2 Further literature review

**Parametric Programming and Sensitivity Analysis.** This stream of work aims to understand how the optimal decision and value change when the underlying problem parameters are perturbed. Sensitivity analysis typically focuses on small, local perturbations, asking how far one can move in a given direction while preserving optimality (Ward and Wendell (1990), Xu and Burer (2017)). Multiparametric programming, by contrast, considers larger, structured changes in the parameters and aims to characterize the full mapping from parameters to optimal solutions, often by partitioning the parameter space into regions where the set of minimizers remains constant (Gal and Nedoma (1972), Saaty and Gass (1954)). The connection to our work lies in the shared goal of studying the interplay between problem parameters and optimal solutions. However, while sensitivity analysis and parametric programming aim to describe how solutions evolve as parameters vary, our focus is on identifying which datasets—i.e., which function parameters—are sufficient to recover the optimal solution.

**Contextual Optimization.** In contextual optimization, as in our setting, a decision-maker aims to choose a decision $x \in \mathcal{X}$ minimizing the loss $L(x, \theta)$, where $\theta$ is unknown (Sadana et al. 2023, Hu et al. 2022, Bertsimas and Kallus 2020). The decision maker also has access to side information $\phi \in \mathbb{R}^p$ that is correlated with $\theta$. Given empirical samples from the joint distribution of $\phi$ and $\theta$, the decision-maker needs to learn a policy $\pi$ that maps side information $\phi$ to an optimal decision $x \in \mathcal{X}$. Within this literature, much of the work—particularly in linear optimization—focuses on constructing data-driven surrogates of the unknown loss function, with the goal of improving decision quality rather than merely predicting losses. A prominent line of research in this direction is the *predict-and-optimize* framework (Elmachtoub and Grigas 2022). This paradigm is similar to ours in the sense that the aim is to directly focus on optimal decisions rather than predictions. However, the fundamental difference between contextual optimization and our setting is that our main concern is to understand *how to select the most informative dataset*, whereas in contextual optimization, the data is already given and one must determine how to use it to produce an optimal decision policy. More recent work in contextual optimization has also considered *adaptive* data-selection strategies (Liu et al. 2023).

**Set-based vs. Distribution-based Modeling of Uncertainty.** In our problem, we chose to model uncertainty through a set ($\theta \in \Theta$) similar to the robust optimization literature. This is in contrast to modeling uncertainty as $\theta$ following some known distribution, as in Bayesian optimization, for example. This modeling choice of uncertainty has been widely discussed in the robust optimization literature (Ben-Tal et al. 2009, Bertsimas et al. 2011, Delage and Ye 2010).

Set-based uncertainty has several practical advantages in our context compared to distribution-based uncertainty. For instance, set-based approaches typically rely on milder and often more realistic assumptions, as they do not require a fully specified probabilistic model of uncertainty. Instead, uncertainty is captured through bounds or confidence sets that are valid for a general class of distributions (such as with a given finite moments, or a given support). This is particularly appealing in settings where the true distribution is unknown, partially observed, or difficult to estimate reliably. Moreover, set-based formulations often lead to more tractable optimization problems and are less sensitive to model misspecification (see Bertsimas et al. (2018), Ben-Tal and Nemirovski (2002)). For instance, in Bayesian Experimental Design, standard approaches require expensive computations to evaluate expected information gains in high-dimensional spaces and are highly sensitive to model misspecification, which can lead to suboptimal results (Rainforth et al. 2024).

## 3 Problem Formulation

We study decision-making tasks modeled as linear programs (LPs). That is where the loss $L(x, \theta) = \theta^\top x$ is linear, and the decision set $\mathcal{X} = \{x \in \mathbb{R}^d, \ Ax = b, \ x \geq 0\}$ is a polyhedron, for $A \in \mathbb{R}^{m \times d}, \ b \in \mathbb{R}^m$. The decision-maker's task is then to solve the LP

$$\min_{x \in \mathcal{X}} c^\top x, \tag{1}$$

where $\mathcal{X}$ is assumed to be bounded. The unknown parameter—or state of nature—here is the cost vector $c$. The decision-maker only knows it to be in some given uncertainty set $\mathcal{C} \subset \mathbb{R}^d$, which captures prior information on $c$ (these are $\theta$ and $\Theta$).

To solve the linear program, the decision-maker can complement their knowledge $c \in \mathcal{C}$ by data on the task. A dataset $\mathcal{D} \subset \mathbb{R}^d$ consists of a set of queries to evaluate the objective function. That is a dataset gives access to the observations $c^\top q$ for $q \in \mathcal{D}$. We focus on the noiseless setting, where each observation $c^\top q$ is exact. This simplification enables a sharp characterization of informativeness. We then show how the core insights naturally extend to noisy observations in Proposition 3.

The fundamental question we seek to address is which datasets are *sufficient* to solve the linear program. We formalize such a property next. Here $\mathcal{P}(\mathcal{X})$ denotes subsets of $\mathcal{X}$.

**Definition 1** (Sufficient Decision Dataset). A set $\mathcal{D} := \{q_1, \ldots, q_N\}$ is a sufficient decision dataset for uncertainty set $\mathcal{C}$ and decision set $\mathcal{X}$ if there exists a mapping $\hat{X} : \mathbb{R}^N \longrightarrow \mathcal{P}(\mathcal{X})$ such that

$$\forall c \in \mathcal{C}, \quad \hat{X}\left(c^\top q_1, \ldots, c^\top q_N\right) = \arg\min_{x \in \mathcal{X}} c^\top x.$$

When there is no ambiguity on $\mathcal{C}$ and $\mathcal{X}$, we simply say that $\mathcal{D}$ is a sufficient decision dataset.

Definition 1 states that a dataset is sufficient if there exists a mapping that can recover the optimal solution of the decision-making task using *only* the dataset's observations and prior information ($c \in \mathcal{C}$).

Naturally, $\mathcal{D} = \{e_1, \ldots, e_d\}$, where $(e_i)_{i \in [d]}$ are canonical basis vectors is a sufficient decision dataset. In fact, observing $c^\top e_i = c_i$ for all $i \in [d]$ amounts to fully observing $c$, and solving the linear program with complete information with $\hat{X}((c^\top q)_{q \in \mathcal{D}}) = \hat{X}(c) := \arg\min_{x \in \mathcal{X}} c^\top x$. The question is then whether there exist other, potentially smaller sufficient datasets. That is, what is the least amount of information required to solve the task? As we will show, whether a dataset is sufficient depends critically on the uncertainty set $\mathcal{C}$ and the feasible region $\mathcal{X}$, since these jointly determine which directions of $c$ affect the optimal decision.

If the goal is to solve the linear program (1), a natural relaxation of Definition 1 is to require only that a dataset permits recovery of *some* optimal solution, rather than the *entire set* of optimal solutions. We show in the next proposition that, under mild structural assumptions, this property is equivalent to the property of Definition 1. This means that any dataset that recovers one solution also recovers all solutions. The proof of this equivalence is nontrivial and relies on several structural results we develop later in the paper.

**Proposition 1** (One vs All Optimal Solutions). *Let $\mathcal{C}$ be an open convex set and $\mathcal{D} := \{q_1, \ldots, q_N\}$ a dataset. The following are equivalent:*

1. *There exists a mapping $\hat{X} : \mathbb{R}^N \longrightarrow \mathcal{P}(\mathcal{X})$ such that $\forall c \in \mathcal{C}, \ \hat{X}\left(c^\top q_1, \ldots, c^\top q_N\right) = \arg\min_{x \in \mathcal{X}} c^\top x$.*

2. *There exists a mapping $\hat{x} : \mathbb{R}^N \longrightarrow \mathcal{X}$ such that $\forall c \in \mathcal{C}, \ \hat{x}\left(c^\top q_1, \ldots, c^\top q_N\right) \in \arg\min_{x \in \mathcal{X}} c^\top x$.*

Notice that observing $c^\top q$ for all $q \in \mathcal{D}$ is equivalent to observing the projection of $c$ onto the span of $\mathcal{D}$. This implies that Definition 1 is equivalent to the following characterization, which gives a valuable perspective. For any subspace $F \subset \mathbb{R}^d$ and $u \in \mathbb{R}^d$, we denote $u_F$ the projection of $u$ in $F$.

**Proposition 2.** $\mathcal{D} := \{q_1, \ldots, q_N\}$ *is a sufficient decision dataset for uncertainty set $\mathcal{C}$ and decision set $\mathcal{X}$ if and only if*

$$\forall c, c' \in \mathcal{C}, \quad c_{\text{span } \mathcal{D}} = c'_{\text{span } \mathcal{D}} \implies \arg\min_{x \in \mathcal{X}} c^\top x = \arg\min_{x \in \mathcal{X}} c'^\top x.$$

In words, Proposition 2 formulates that a dataset $\mathcal{D}$ is sufficient if any two cost vectors that are equivalent from the perspective of the information provided by $\mathcal{D}$ (and $\mathcal{C}$) lead to the same optimal solutions in the decision-making problem.

This characterization suggests a natural algorithm for solving the LP (1) when given a sufficient dataset $\mathcal{D} = \{q_1, \ldots, q_N\}$. Suppose we observe values $o_i = c^\top q_i, i \in [N]$ for an unknown cost

vector $c \in \mathcal{C}$. We then compute $\hat{c} \in \arg\min\{\sum_{i=1}^{N}(c'^{\top}q_i - o_i)^2 \; : \; c' \in \mathcal{C}\}$ and use $\hat{c}$ to solve the decision problem $\min_{x \in \mathcal{X}} \hat{c}^{\top}x$. This procedure recovers the projection of $c$ onto $\text{span}\,\mathcal{D}$ while respecting the prior of $\mathcal{C}$. This ensures $\hat{c}_{\text{span}\,\mathcal{D}} = c_{\text{span}\,\mathcal{D}}$ as $c \in \mathcal{C}$, and since the dataset is sufficient, guarantees that the resulting decision is task-optimal (Proposition 2).

When the observations are noisy, a sufficient dataset can still yield a correct decision. In particular, estimating an approximate cost vector $\hat{c}$ via least-squares from noisy observations still leads to an optimal decision, as long as the noise is sufficiently small.

**Proposition 3** (Noisy Observations). *Let $\mathcal{C} \subset \mathbb{R}^d$ be an open set, and $\mathcal{D} := \{q_1, \ldots, q_r\}$ a sufficient decision dataset for $\mathcal{C}$. Let $c \in \mathcal{C}$. Let $\varepsilon_1, \ldots, \varepsilon_r \in \mathbb{R}$, and for all $i \in [r]$, $o_i = c^{\top}q_i + \varepsilon_i$. Let $\hat{c} \in \arg\min\{\sum_{i=1}^{r}(c'^{\top}q_i - o_i)^2 \; : \; c' \in \mathcal{C}\}$. There exists $\kappa > 0$ such that if $\|\varepsilon\| < \kappa$, then $\arg\min_{x \in \mathcal{X}} \hat{c}^{\top}x \subset \arg\min_{x \in \mathcal{X}} c^{\top}x$.*

# 4 Characterizing Sufficient Datasets

Given an uncertainty set $\mathcal{C}$ and a decision set $\mathcal{X}$, we would like to characterize sufficient decision datasets and eventually construct such datasets. As in Blackwell's theory, the difficulty of such characterizations depends on the richness of the uncertainty set $\mathcal{C}$. In fact, the first results by Blackwell (1949, 1951) and Sherman (1951) were for a set $\mathcal{C}$ with only two elements. That is, the data needs to distinguish only two alternative states of nature. The result was later extended to the finite sets by Blackwell (1953) and then to infinite sets with regularity conditions by Boll (1955).

## 4.1 Characterization Under No Prior Knowledge

We begin with the case of no prior knowledge, i.e., $\mathcal{C} = \mathbb{R}^d$, which isolates how the structure of the decision set $\mathcal{X}$ alone determines what information is necessary to recover the optimal solution. We will then study the case of convex sets. To formulate our result, define $F_0 = \text{span}\,\{e_i, \; i \in [d], \; \exists x \in \mathcal{X}, \; x_i \neq 0\}$ where $e_i$ is the $i-$th element of the canonical basis. $F_0$ captures the coordinates that can take non-zero values in feasible solutions of $\mathcal{X}$. That is $F_0^{\perp}$ captures coordinates that are identically zero in all feasible solutions: $e_i \in F_0^{\perp} \implies \forall x \in \mathcal{X}, \; x_i = 0$.

**Proposition 4.** *Suppose $\mathcal{C} = \mathbb{R}^d$. $\mathcal{D}$ is a sufficient decision dataset if and only if $F_0 \cap \text{Ker}\,A \subset \text{span}\,\mathcal{D}$. Furthermore, when the condition $F_0 \cap \text{Ker}\,A \subset \text{span}\,\mathcal{D}$ is not satisfied, for any mapping $\hat{x} : \mathbb{R}^N \longrightarrow \mathcal{X}$, and any $K > 0$, there exists $c \in \mathbb{R}^d$ such that $c^{\top}\hat{x}\left(c^{\top}q_1, \ldots, c^{\top}q_N\right) \geq K + \min_{x \in \mathcal{X}} c^{\top}x$.*

Proposition 4 indicates that, already with no prior knowledge, not all the information on $c$ is required to solve the optimization problem. In fact, the dataset needs only to capture "relevant" information for the decision-making task, defined by $\mathcal{X}$. The proposition shows that these are the directions in the null space of $A$ ($\text{Ker}\,A$), that act on active variables ($F_0$).

Let us provide an intuitive explanation for this result. Since every $x \in \mathcal{X}$ satisfies $x_i = 0$ for all $e_i \in F_0^{\perp}$, the components of $c$ along $F_0^{\perp}$ do not influence the objective and the dataset $\mathcal{D}$ need not capture these directions. Hence, we can, without loss of generality, restrict attention to the subspace $F_0$ and replace the variable $x$ with its projection $x_{F_0}$. The objective function can be decomposed as $x \longmapsto c_{\text{Ker}\,A}^{\top}x + c_{(\text{Ker}\,A)^{\perp}}^{\top}x$. Because $\mathcal{X}$ lies in an affine space parallel to $\text{Ker}\,A$, any change from a feasible decision $x$ to another feasible decision $x + \delta$ necessarily verifies $\delta \in \text{Ker}\,A$. Therefore, $c_{(\text{Ker}\,A)^{\perp}}^{\top}x$ is constant across all feasible solutions. This means that only the projection of $c$ in $\text{Ker}\,A$ matters when comparing costs of feasible decisions.

The second part of Proposition 4 formalizes a dichotomy: either the dataset is sufficient and enables optimal decision recovery, or any algorithm may incur arbitrarily large suboptimality in the worst case. This sharp divide, however, is specific to the unstructured case $\mathcal{C} = \mathbb{R}^d$. As we will see next, imposing structure on $\mathcal{C}$ significantly enriches the notion of sufficiency.

## 4.2 Characterization under Convex Uncertainty Sets

The goal now is to characterize sufficient datasets for any convex uncertainty set $\mathcal{C}$. We start by introducing some geometric notions that are useful to understand the sufficiency of a dataset.

**Definition 2** (Extreme Points). An element $x \in \mathcal{X}$ is an extreme point if and only if there are no $\lambda \in (0,1)$ and $y, z \in \mathcal{X}$ such that $x = \lambda y + (1-\lambda)z$. The set of extreme points of $\mathcal{X}$ is denoted $\mathcal{X}^{\angle}$.

From every extreme point, there is a set of *feasible directions* that allow changing the solution while remaining in the polyhedron $\mathcal{X}$—the feasible region. Out of these feasible directions, *extreme directions* allow moving to "neighboring" extreme points.

**Proposition 5** (Feasible and Extreme Directions). *For every $x^{\star} \in \mathcal{X}^{\angle}$, we denote*

$$\mathrm{FD}(x^{\star}) = \{\delta \in \mathbb{R}^d, \ \exists \varepsilon > 0, \ x^{\star} + \varepsilon \delta \in \mathcal{X}\}$$

*the set of feasible directions from $x^{\star}$ in $\mathcal{X}$. $\mathrm{FD}(x^{\star})$ is a polyhedral cone and $\mathrm{FD}(x^{\star}) \subset F_0 \cap \mathrm{Ker}\, A$. We denote $D(x^{\star})$ the set of extreme directions of $\mathrm{FD}(x^{\star})$: non-zero vectors in $\mathrm{FD}(x^{\star})$ that cannot be written as a convex combination of two non-proportional elements of $\mathrm{FD}(x^{\star})$.*

In linear programs, optimal solutions are attained in extreme points $\mathcal{X}^{\angle}$. Every extreme point is associated with a set of cost vectors $c$ for which it is optimal. This set forms a cone, as illustrated in Fig. 1 (middle).

**Proposition 6** (Optimality Cones). *For every $x^{\star} \in \mathcal{X}^{\angle}$, we denote $\Lambda(x^{\star}) = \{c \in \mathbb{R}^d \ : \ x^{\star} \in \arg\min_{x \in \mathcal{X}} c^{\top} x\}$. We have $\Lambda(x^{\star}) = \{c \in \mathbb{R}^d, \ \forall \delta \in D(x^{\star}), \ c^{\top}\delta \geq 0\}$. For every $\delta \in D(x^{\star})$, we denote $F(x^{\star}, \delta) := \Lambda(x^{\star}) \cap \{\delta\}^{\perp}$ the face of the cone $\Lambda(x^{\star})$ that is perpendicular to $\delta$. Furthermore, $\Lambda(x^{\star})$ is the dual cone of $\mathrm{FD}(x^{\star})$.*

Notice that since $\mathcal{X}$ is bounded, for any $c \in \mathbb{R}^d$, there exists $x^{\star} \in \mathcal{X}^{\angle}$ such that $c \in \Lambda(x^{\star})$, and consequently $\mathbb{R}^d = \bigcup_{x^{\star} \in \mathcal{X}^{\angle}} \Lambda(x^{\star})$ as illustrated in Fig. 1 (middle). Neighboring cones share boundaries corresponding to their faces (Fig. 1, right), where multiple solutions can be optimal.

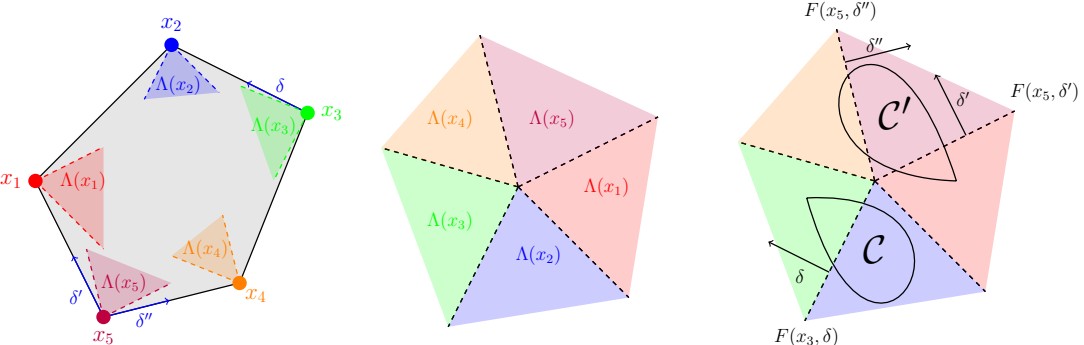

Figure 1: Optimality cones relative to $\mathcal{X}$ (left), relative to the origin (middle) and examples of the uncertainty sets ($\mathcal{C}$ and $\mathcal{C}'$) relative to the optimality cones (right).

With the notion of optimality cones, solving a linear program for a given cost vector $c$ amounts to finding to which optimality cone it belongs. A dataset is therefore sufficient if it enables to determine the optimality cone of each possible $c \in \mathcal{C}$. As $\mathcal{C}$ already restricts the location of its cost vectors, our data only needs to discriminate between cones overlapping with $\mathcal{C}$ as illustrated in Fig. 1 (right).

To provide further intuition, consider the example shown in Fig. 1 (right). The set $\mathcal{C}$ intersects only the cones $\Lambda(x_2)$ (blue) and $\Lambda(x_3)$ (green), hence, the cost vectors can only be in these two cones. Clearly, observing their projection on the span of $\delta$ is sufficient to determine which of the two cones they belong to. The set $\mathcal{C}'$, however, intersecting $\Lambda(x_4), \Lambda(x_5)$ and $\Lambda(x_1)$, requires projections on the span of both $\delta'$ and $\delta''$. These vectors are not arbitrary; these are extreme directions that move from one cone to its adjacent cone, inducing a face where both cones intersect. The illustration highlights that such vectors are necessary to capture by our data when the face they induce intersects the uncertainty set $\mathcal{C}$. Hence, it is natural to introduce the following set of *relevant extreme directions*.

**Definition 3** (Relevant Extreme Directions). Given $\mathcal{C} \subset \mathbb{R}^d$, we define

$$\Delta(\mathcal{X}, \mathcal{C}) = \{\delta \in \mathbb{R}^d \ : \ \exists x^{\star} \in \mathcal{X}^{\angle}, \ \delta \in D(x^{\star}) \text{ and } F(x^{\star}, \delta) \cap \mathcal{C} \neq \varnothing\}.$$

In the illustration of Fig. 1, we have $\mathrm{span}\, \Delta(\mathcal{X}, \mathcal{C}) = \mathrm{span}\, \{\delta\}$ and $\mathrm{span}\, \Delta(\mathcal{X}, \mathcal{C}') = \mathrm{span}\, \{\delta', \delta''\}$, and it is necessary to observe the projections on $\Delta(\mathcal{X}, \mathcal{C})$ and $\Delta(\mathcal{X}, \mathcal{C}')$ to recover optimal solutions for uncertainty sets $\mathcal{C}$ and $\mathcal{C}'$ respectively. This leads to our first main theorem.

**Theorem 1.** *Let $\mathcal{C}$ be an open convex set. $\mathcal{D}$ is a sufficient decision dataset for uncertainty set $\mathcal{C}$ and decision set $\mathcal{X}$ if and only if $\Delta(\mathcal{X}, \mathcal{C}) \subset \operatorname{span} \mathcal{D}$.*

Theorem 1 is a fundamental characterization of sufficiency, by what information the dataset needs to capture relative to the prior knowledge $\mathcal{C}$ and the problem structure $\mathcal{X}$. The result also indicates that such a minimal dataset is, in general, not unique. A careful reader might remark that Theorem 1 should imply Proposition 4 when $\mathcal{C} = \mathbb{R}^d$. In fact, $\Delta(\mathcal{X}, \mathbb{R}^d)$ is the set of all extreme directions of the polyhedron, which indeed precisely spans $\operatorname{Ker} A \cap F_0$. Finally, we remark that in the proof of the result, only convexity is required for sufficiency, while only openness is required for necessity.

### 4.3 An Algorithmically Tractable Characterization

We now develop a second characterization of dataset sufficiency that is particularly well-suited to algorithmic construction.

The set $\Delta(\mathcal{X}, \mathcal{C})$ of relevant extreme directions of Theorem 1 can be seen intuitively as the set of differences $x_1 - x_2$ of *neighboring* extreme points $x_1, x_2 \in \mathcal{X}^{\angle}$, that are optimal for some $c \in \mathcal{C}$. By relaxing the "neighboring" condition and optimality for a common $c \in \mathcal{C}$, we arrive at a broader set of directions induced by all pairs of optimal extreme points—which we call *reachable solutions*.

**Definition 4** (Reachable Solutions). Given $\mathcal{C} \subset \mathbb{R}^d$, we define

$$
\mathcal{X}^{\star}(\mathcal{C}) := \left\{ x^{\star} \in \mathcal{X}^{\angle}, \ \exists c \in \mathcal{C}, \ x^{\star} \in \arg\min_{x \in \mathcal{X}} c^{\top} x \right\} = \bigcup_{c \in \mathcal{C}} \arg\min_{x \in \mathcal{X}} c^{\top} x.
$$

and its set of directions as $\operatorname{dir}(\mathcal{X}^{\star}(\mathcal{C})) := \operatorname{span} \{ x_1 - x_2 \ : \ x_1, x_2 \in \mathcal{X}^{\star}(\mathcal{C}) \}$.

The set $\operatorname{dir}(\mathcal{X}^{\star}(\mathcal{C}))$ is equal to the span of the set of differences between *any* two elements $x_1, x_2 \in \mathcal{X}$ such that there exists $c_1, c_2 \in \mathcal{C}$ such that $x_1 \in \arg\min_{x \in \mathcal{X}} c_1^{\top} x$ and $x_2 \in \arg\min_{x \in \mathcal{X}} c_2^{\top} x$. By construction, we have $\operatorname{span} \Delta(\mathcal{X}, \mathcal{C}) \subset \operatorname{dir}(\mathcal{X}^{\star}(\mathcal{C}))$ since each relevant extreme direction corresponds to a direction between optimal solutions. The following theorem shows that these quantities are indeed equal.

**Theorem 2.** *For any convex set $\mathcal{C} \subset \mathbb{R}^d$, we have $\operatorname{span} \Delta(\mathcal{X}, \mathcal{C}) = \operatorname{dir}(\mathcal{X}^{\star}(\mathcal{C}))$.*

The converse inclusion proven in this theorem is not immediate. In fact, for a general polyhedron $\mathcal{X}$ and $\mathcal{C}$ (see Fig. 1 with $\mathcal{C}'$ for eg.), $\Delta(\mathcal{X}, \mathcal{C})$ is much smaller than the set of differences of elements in $\mathcal{X}^{\star}(\mathcal{C})$ but their spans are equal. To prove the equality, we prove that for any $x, x' \in \mathcal{X}^{\star}(\mathcal{C}) \cap \mathcal{X}^{\angle}$, there exists a sequence of extreme points $x_1, \ldots, x_h \in \mathcal{X}^{\star}(\mathcal{C})$ such that $x_1 = x$ and $x_h = x'$ and for any $i \in [h-1]$, $x_{i+1} - x_i \in \Delta(\mathcal{X}, \mathcal{C})$. In other words, $x_{i+1}, x_i$ are neighbors and are both optimal for some $c_i \in \mathcal{C}$. This implies that every element in $\operatorname{dir}(\mathcal{X}^{\star}(\mathcal{C}))$ can be written as a finite linear combination of elements in $\Delta(\mathcal{X}, \mathcal{C})$, completing the equality. Relating again to Proposition 4 of the case $\mathcal{C} = \mathbb{R}^d$, careful linear algebra shows that indeed $\operatorname{dir}(\mathcal{X}^{\star}(\mathbb{R}^d)) = \operatorname{dir}(\mathcal{X}) = \operatorname{Ker} A \cap F_0$.

Theorem 1 implies that to construct a sufficient decision dataset it suffices to find a basis of $\operatorname{dir}(\mathcal{X}^{\star}(\mathcal{C}))$ rather than $\operatorname{span} \Delta(\mathcal{X}, \mathcal{C})$, which is a much simpler task. The following corollary will indeed be the basis of our algorithm in the next section.

**Corollary 1.** *Let $\mathcal{C}$ be an open convex set. $\mathcal{D}$ is a sufficient decision dataset for $\mathcal{C}$ if and only if $\operatorname{dir}(\mathcal{X}^{\star}(\mathcal{C})) \subset \operatorname{span} \mathcal{D}$.*

## 5 A Data Collection Algorithm: Finding Minimal Sufficient Datasets

We now turn to the practical problem of selecting a minimal—i.e., smallest or least costly—dataset $\mathcal{D}$ that enables generalization from prior contextual knowledge (captured by $c \in \mathcal{C}$) to a specific decision-making task (defined by $\mathcal{X}$).

In many practical settings, data collection is subject to constraints on what can be queried. We model this by restricting the dataset to lie in a predefined query set $\mathcal{Q} \subset \mathbb{R}^d$, so that $\mathcal{D} \subseteq \mathcal{Q}$. For example, in the hiring problem discussed in Section 1, $\mathcal{Q}$ is the set of canonical basis vectors—a data point is interviewing one candidate. Corollary 1 implies then that the data collection problem becomes: finding the smallest $\mathcal{D} \subset \mathcal{Q}$ verifying $\operatorname{dir}(\mathcal{X}^{\star}(\mathcal{C})) \subset \operatorname{span} \mathcal{D}$.

We will focus in what follows on the important case of $\mathcal{Q}$ being the set of canonical basis vectors. That is, each query in the data consists in evaluating one coordinate of unknown cost vector $c$, which represents the score of some candidate. In this case, given $\mathrm{dir}\,(\mathcal{X}^\star\,(\mathcal{C}))$, represented by a basis $v_1, \ldots, v_k$, it is clear that the smallest sufficient data set, verifying the spanning condition of Corollary 1, is $\mathcal{D} = \{e_i\ :\ i \in [d],\ \exists j \in [k],\ v_j^\top e_i \neq 0\}$. This is all the non-zero coordinates of basis vectors of $\mathrm{dir}\,(\mathcal{X}^\star\,(\mathcal{C}))$, which are required to span $\mathrm{dir}\,(\mathcal{X}^\star\,(\mathcal{C}))$. This case can be generalized in a straightforward manner to the case where $\mathcal{Q}$ is any basis of $\mathbb{R}^d$; see Appendix B.2.

The central step in this approach is therefore to compute $\mathrm{dir}\,(\mathcal{X}^\star\,(\mathcal{C}))$ and construct a basis for it, which is the focus of the remainder of this section. We can write $\mathrm{dir}\,(\mathcal{X}^\star\,(\mathcal{C})) = \mathrm{span}\,\{x_0 - x,\ x \in \mathcal{X}^\star\,(\mathcal{C})\}$ for some $x_0 \in \mathcal{X}^\star\,(\mathcal{C})$. Hence, to compute $\mathrm{dir}\,(\mathcal{X}^\star\,(\mathcal{C}))$, we can iteratively add elements of it while ensuring we increase the dimension at every step. This is formalized in Algorithm 1.

---

**Algorithm 1** Meta-Algorithm Computing $\mathrm{dir}\,(\mathcal{X}^\star\,(\mathcal{C}))$

---

**Input:** Decision set $\mathcal{X}$, Uncertainty set $\mathcal{C}$.
**Output:** A basis $\mathcal{D} \subset \mathbb{R}^d$ of $\mathrm{dir}\,(\mathcal{X}^\star\,(\mathcal{C}))$.
Initialize $\mathcal{D}$ to $\varnothing$.
Set $x_0 \in \arg\min_{x \in \mathcal{X}} c_0^\top x$ for some $c_0 \in \mathcal{C}$.
**while** there exists $c \in \mathcal{C}$, $x^\star \in \arg\min_{x \in \mathcal{X}} c^\top x$ such that $x^\star - x_0 \notin \mathrm{span}\,\mathcal{D}$.
$\quad \mathcal{D} \leftarrow \mathcal{D} \cup \{x^\star - x_0\}$.
**return** $\mathcal{D}$

---

The main step in Algorithm 1 (condition of the while loop) can be seen as verifying whether the optimization problem

$$\sup\{\,\|\mathrm{proj}_{(\mathrm{span}\,\mathcal{D})^\perp}\,(x^\star - x_0)\|\ :\ c \in \mathcal{C},\ x^\star \in \arg\min_{x \in \mathcal{X}} c^\top x\}, \tag{2}$$

where $\mathrm{proj}_{(\mathrm{span}\,\mathcal{D})^\perp}\,(\cdot)$ is the projection map onto $(\mathrm{span}\,\mathcal{D})^\perp$, has a solution with a non-zero objective. This optimization problem has two main challenges: first, it entails the inherently difficult task of maximizing a convex objective, and second, it has a bilinear, bi-level constraint $x^\star \in \arg\min_{x \in \mathcal{X}} c^\top x$ as both $c$ and $x^\star$ are variables and $x^\star$ must be an optimal solution to a linear program parameterized by c.

**Linearizing the objective.** Remark that if $\alpha$ is a randomly generated Gaussian vector, then any vector $v$, with $\|v\| > 0$, satisfies $\mathrm{Prob}(\alpha^\top v = 0) = 0$. Hence, if Problem (2) admits a solution $\bar{x}$ verifying $\|\mathrm{proj}_{(\mathrm{span}\,\mathcal{D})^\perp}\,(\bar{x} - x_0)\| > 0$, then $\alpha^\top \mathrm{proj}_{(\mathrm{span}\,\mathcal{D})^\perp}\,(\bar{x} - x_0) \neq 0$ with probability 1, and therefore either maximizing or minimizing $\alpha^\top \mathrm{proj}_{(\mathrm{span}\,\mathcal{D})^\perp}\,(x^\star - x_0)$ must lead to a non-zero objective with probability 1. This is a linear objective as the projection onto a subspace is linear.

**Linearizing the bilinear, bilevel constraint.** To address the second challenge, we use complementary slackness conditions, which characterize the optimal solutions of linear programs. We replace $x^\star \in \arg\min_{x \in \mathcal{X}} c^\top x,\ c \in \mathcal{C}$ by

$$x^\star \geq 0,\ s \geq 0,\ \lambda \in \mathbb{R}^m,\ c \in \mathcal{C},$$
$$Ax^\star = b,\ A^\top \lambda + s = c,\ x_i^\star s_i = 0,\ \forall i \in [d]$$

The bilinear constraint $x_i^\star s_i = 0$ can be linearized by introducing a binary variable $\tau_i \in \{0, 1\}$ and adding the constraint $1 - \epsilon s_i \geq \tau_i \geq \epsilon x_i^\star$ with $\epsilon > 0$ a small constant. When $\mathcal{C}$ is a polyhedron, the resulting formulation is a mixed-integer linear program (MILP) with linear constraints and objectives.

Putting everything together gives Algorithm 2 for linear programs, which is detailed in Appendix B. The algorithm will terminate in exactly $\dim\,\mathrm{dir}\,(\mathcal{X}^\star\,(\mathcal{C}))$ iterations. When $\mathcal{C}$ is a polyhedron, each iteration involves solving a mixed integer program with $O(d+m)$ variables and $O(d+m+\mathrm{constr}(\mathcal{C}))$ constraints where $\mathrm{constr}(\mathcal{C})$ is the number of constraints defining $\mathcal{C}$.

**Theorem 3** (Correctness). *Algorithm 2 terminates with probability 1 after* $\dim\,\mathrm{dir}\,(\mathcal{X}^\star\,(\mathcal{C})) \leq d$ *steps and outputs a basis of* $\mathrm{dir}\,(\mathcal{X}^\star\,(\mathcal{C}))$.

## 6   Application: Hiring Interviews

To illustrate our insights, we apply our theoretical framework to the hiring problem detailed in Section 1. The smallest sufficient dataset here is the smallest subset of candidates to interview

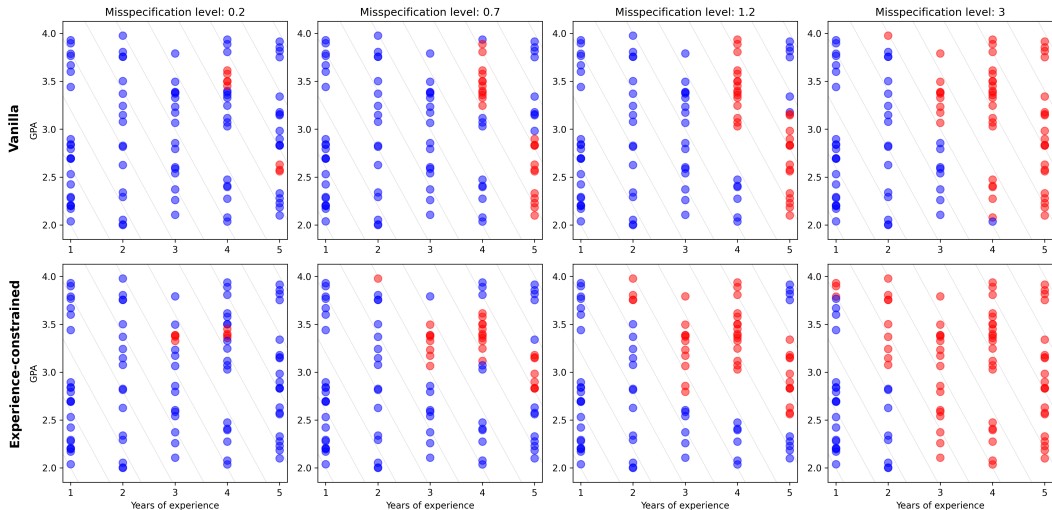

Figure 2: Candidates to be interviewed (in red) to make an optimal hiring decision. Number of candidates to interview from left to right for top and bottom row respectively: $8, 24, 31, 52$ and $8, 28, 43, 70$.

to recover the optimal hiring decision. The application illustrates how the task constraints and uncertainty set shape data needs. The goal is to hire 20 candidates from a pool of $d = 100$ candidates. Each candidate is associated with two features: GPA and years of experience. We study two settings: vanilla hiring, with only a total hire cap, and experience-constrained hiring, which also limits hires per seniority group. The decision sets are $\mathcal{X}_{\text{vanilla}} := \{x \in \{0, 1\}^d : \sum_{i=1}^d x_i \leq 20\}$ and $\mathcal{X}_{\text{experience}} := \{x \in \mathcal{X}_{\text{vanilla}} : \forall j \in [4], \sum_{i \in I_j} x_i \leq 8\}$, where $I_j$ is the set of candidates with $j$ years of experience. These constraints are totally unimodular, so relaxing $x \in \{0, 1\}^d$ to $x \in [0, 1]^d$ still yields optimal solutions via LP (Wolsey 2020, Chapter 3). We assume a misspecified linear model, i.e. candidate scores belong to

$$\mathcal{C} := \{c \in \mathbb{R}^d : \exists \alpha \in \mathbb{R}^2, \ \exists \varepsilon \in [-\eta, \eta], \ \ell \leq \alpha \leq u, \ c = \alpha^\top \phi + \varepsilon\},$$

where $\eta \geq 0$ controls the misspecification level, and $\ell = (4, 4), u = (5, 5)$. $\phi$ is a feature matrix whose rows are GPAs and years of experience of candidates. Fig. 2 indicates candidates to interview to enable an optimal hiring decision. **Impact of $\mathcal{C}$:** As misspecification increases ($\mathcal{C}$ grows larger), so does the number of required interviews: more uncertainty requires more data points. **Impact of $\mathcal{X}$:** In the first row of Fig. 2, candidates fall into three groups: low scorers (never hired), high scorers (always hired), and mid scorers (interviewed)—an intuitive pattern given the task, automatically recovered by our algorithm. When adding group hiring constraints—second row of Fig. 2— a similar pattern arises, but now across experience groups rather than the entire population: low misspecification yields separate treatment between experience groups, as scores don't overlap across experience levels; high misspecification leads to cross-experience group comparisons and mixing—again, an intuitive pattern given the new constraints. Further discussion about the experiments is available in Appendix D.

## 7 Conclusion and Limitations

This paper introduces a framework for quantifying the informativeness of datasets in decision-making tasks. While our analysis yields sharp results, several natural extensions remain. First, we restrict attention to linear optimization; extending the framework to other problem classes, such as mixed-integer or convex programs, is an important direction even at the cost of approximate characterizations. Second, we assume query sets are basis vectors; accounting for general queries is a hard problem, but opens rich avenues for exploration. Third, our focus on convex, open uncertainty sets excludes important structured cases such as low-dimensional or discrete sets encoding symmetry or logical constraints. Finally, alternative notions of informativeness, such as approximate rather than exact optimality, or noisy observations rather than clean, merit further study.

## Acknowledgments and Disclosure of Funding

This work was partially supported by MIT CGC project "Preparing for a New World of Weather and Climate Extreme", and AFOSR Grant FA9550-23-1-0190.

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

# A    Proofs

## A.1    Proof of Proposition 2

*Proof.*

- ($\Rightarrow$) Assume that $\mathcal{D}$ is a sufficient decision dataset. Let $c, c' \in \mathcal{C}$ such that $c_{\text{span } \mathcal{D}} = c'_{\text{span } \mathcal{D}}$. We have for any $q \in \mathcal{D}$, $c^\top q = c'^\top q$. Let $\hat{X}$ given by Definition 1. We have $\hat{X}\left(c^\top q_1, \dots, c^\top q_N\right) = \hat{X}\left(c'^\top q_1, \dots, c'^\top q_N\right)$ i.e. $\arg\min_{x \in \mathcal{X}} c^\top x = \arg\min_{x \in \mathcal{X}} c'^\top x$.

- ($\Leftarrow$) Assume that $\mathcal{D}$ satisfies the property of the proposition. Since for any $c, c' \in \mathcal{C}$ we have $c_{\text{span } \mathcal{D}} = c'_{\text{span } \mathcal{D}} \iff (c^\top q)_{q \in \mathcal{D}} = (c'^\top q)_{q \in \mathcal{D}}$, then for any $c \in \mathcal{C}$, we define $\hat{X}\left(c^\top q_1, \dots, c^\top q_N\right)$ to be equal to $\arg\min_{x \in \mathcal{X}} c'^\top x$ for any $c'$ such that $c'_{\text{span } \mathcal{D}} = c_{\text{span } \mathcal{D}}$. This mapping is well-defined and verifies the desired property.

$\square$

## A.2    Proof of Proposition 3

*Proof.* Let $Q$ be a matrix whose rows are the elements of $\mathcal{D}$ and $\hat{c}(o_1, \dots, o_r) \in \arg\min\{\sum_{i=1}^r (c'^\top q_i - o_i)^2 : c' \in \mathcal{C}\}$. Let $\eta := Q\hat{c}(o_1, \dots, o_r) - Qc_{\text{true}}$. Since $\hat{c}(o_1, \dots, o_r) = \arg\min_{c \in \mathcal{C}} \|Qc - o\|$, then we have $\|Q\hat{c}(o_1, \dots, o_r) - Qc_{\text{true}} - \varepsilon\| \leq \|\varepsilon\|$. Hence, we have $\|Q\hat{c}(o_1, \dots, o_r) - Qc_{\text{true}}\| \leq 2\|\varepsilon\|$ and consequently $\|\eta\| \leq 2\|\varepsilon\|$. We would like to show that the distance between the projections of $c_{\text{true}}$ and $\hat{c}(o_1, \dots, o_r)$ in the span of $\mathcal{D}$ is upper bounded by $O(\|\varepsilon\|)$. Consider $\alpha_{\text{true}}, \hat{\alpha} \in \mathbb{R}^r$ such that $\hat{c}(o_1, \dots, o_r)_{\text{span } \mathcal{D}} = Q^\top \hat{\alpha}$ and $c_{\text{true},\text{span } \mathcal{D}} = Q^\top \alpha_{\text{true}}$. Without loss of generality, we can assume that $\mathcal{D}$ is linearly independent. Indeed, if $\mathcal{D}$ was linearly dependent, it would provide exactly the same information as any smallest cardinality subset of $\mathcal{D}$ that spans all elements of $\mathcal{D}$. In this case $Q$ is full rank and $QQ^\top$ is invertible. We have

$$Q\hat{c}(o_1, \dots, o_r) - Qc_{\text{true}} = \eta \implies Q(\hat{c}(o_1, \dots, o_r)_{\text{span } \mathcal{D}} - c_{\text{true},\text{span } \mathcal{D}}) = \eta$$
$$\implies QQ^\top(\hat{\alpha} - \alpha_{\text{true}}) = \eta$$
$$\implies \hat{\alpha} - \alpha_{\text{true}} = (QQ^\top)^{-1}\eta$$
$$\implies \hat{c}(o_1, \dots, o_r)_{\text{span } \mathcal{D}} - c_{\text{true},\text{span } \mathcal{D}} = Q^\top(QQ^\top)^{-1}\eta.$$

Let $U \in \mathbb{R}^{r \times r}$, $V \in \mathbb{R}^{d \times d}$ and $\Sigma \in \mathbb{R}^{r \times d}$ such that $U, V$ are orthogonal matrices and for all $(i, j) \in [r] \times [d]$

$$\Sigma_{ij} = \begin{cases} \sigma_i \text{ the } i-\text{th singular value of } Q & \text{if } i = j \\ 0 & \text{else,} \end{cases}$$

and $Q = U\Sigma V^\top$. We have

$$Q^\top(QQ^\top)^{-1} = U\Sigma V^\top(U\Sigma V^\top V\Sigma^\top U^\top)^{-1}$$
$$= U\Sigma V^\top(U\Sigma\Sigma^\top U^\top)^{-1}$$
$$= V\Sigma^\top U^\top U(\Sigma\Sigma^\top)^{-1}U^\top$$
$$= V\Sigma^\top(\Sigma\Sigma^\top)^{-1}U^\top = V\Sigma'U^\top,$$

where $\Sigma' \in \mathbb{R}^{d \times r}$ satisfies

$$\Sigma'_{ij} = \begin{cases} \frac{1}{\sigma_i} \text{ the } i-\text{th singular value of } Q & \text{if } i = j \\ 0 & \text{else.} \end{cases}$$

Let $\lambda_{\min}(D)$ the smallest singular value of $Q$. The calculations above gives, when $\|.\|$ is the $L^2$ norm,

$$\|c_{\text{true},\text{span } \mathcal{D}} - \hat{c}(o_1, \dots, o_r)_{\text{span } \mathcal{D}}\| = \|Q^\top(QQ^\top)^{-1}\eta\| \leq \|Q^\top(QQ^\top)^{-1}\| \cdot \|\eta\| \leq \frac{2}{\lambda_{\min}(D)}\|\varepsilon\|.$$

We now provide an essential lemma.

**Lemma 1.** *Assume that $\mathcal{C}$ is open. Let $\mathcal{D}$ a sufficient decision dataset for $\mathcal{C} \subset \mathbb{R}^d$. Let $c \in \mathcal{C}$. There exists $\mu > 0$ such that for any $c' \in \mathcal{C}$ such that $\left\| c_{\mathrm{span}\ \mathcal{D}} - c'_{\mathrm{span}\ \mathcal{D}} \right\| < \mu$, we have $\arg\min_{x \in \mathcal{X}} c'^\top x \subset \arg\min_{x \in \mathcal{X}} c^\top x$.*

*Proof.* We assume without loss of generality that $\mathcal{C}$ is compact (it suffices to replace $\mathcal{C}$ by some closed ball of small radius centered around $c$ that is a subset of $\mathcal{C}$). Assume that the result does not hold, i.e. there exists a sequence $c'_n \in \mathcal{C}$ such that $c'_{n,\mathrm{span}\ D}$ converges to $c_{\mathrm{span}\ D}$, and for all $n \in \mathbb{N}$, there exists $x' \in \mathcal{X}^{\angle}$ such that $x' \in \arg\min_{x \in \mathcal{X}} c'^\top_n x \setminus \arg\min_{x \in \mathcal{X}} c^\top x$. Since the number of extreme points in $\mathcal{X}$ are finite, there exists $x' \in \mathcal{X}^{\angle}$ and a strictly increasing map $\varphi : \mathbb{N} \longmapsto \mathbb{N}$ such that for all $n \in \mathbb{N}$, we have $x' \in \arg\min_{x \in \mathcal{X}} c'^\top_{\varphi(n)} x \setminus \arg\min_{x \in \mathcal{X}} c^\top x$. Since $\mathcal{C}$ is compact, we can assume without loss of generality that the sequence $c'_{\varphi(n)}$ is convergent to some $c' \in \mathcal{C}$ (it suffices to extract another time a converging sequence from $c'_{\varphi(n)}$). Consequently, since for all $n \in \mathbb{N}$, $c'_{\varphi(n)} \in \Lambda(x')$ (see Proposition 6 for definition of $\Lambda(x')$), and $\Lambda(x')$ is closed, then $c' \in \Lambda(x')$. Furthermore, we have $c \notin \Lambda(x)$, and $c_{\mathrm{span}\ \mathcal{D}} = c'_{\mathrm{span}\ \mathcal{D}}$, which means that $\arg\min_{x \in \mathcal{X}} c^\top x = \arg\min_{x \in \mathcal{X}} c'^\top x$. This implies that $x' \in \arg\min_{x \in \mathcal{X}} c^\top x$, i.e. $c \in \Lambda(x)$ which is impossible. $\square$

When $\|\varepsilon\| \leq \frac{\mu \lambda_{\min}(D)}{2}$, we have

$$\left\| c_{\mathrm{true,span}\ \mathcal{D}} - \hat{c}(o_1, \ldots, o_r)_{\mathrm{span}\ \mathcal{D}} \right\| < \mu,$$

i.e. from the lemma above, $\arg\min_{x \in \mathcal{X}} \hat{c}(o_1, \ldots, o_r)^\top x \subset \arg\min_{x \in \mathcal{X}} c^\top_{\mathrm{true}} x$.

$\square$

### A.3 Proof of Proposition 4

*Proof.* We denote $F := \mathrm{span}\ \mathcal{D}$. The condition $F_0 \cap \mathrm{Ker}\ A \subset \mathrm{span}\ \mathcal{D}$ is equivalent to $F_0 \cap \mathrm{Ker}\ A \perp F^\perp$, so in order to prove the equivalence with the 3rd proposition, we will prove the equivalence with $F_0 \cap \mathrm{Ker}\ A \perp F^\perp$.

- Assume that $F^\perp \perp F_0 \cap \mathrm{Ker}\ A$. Let $c, c' \in \mathbb{R}^d$ such that $c_F = c'_F$. We will show that they have the same $\arg\min$, which proves sufficient as a result of Proposition 2. We show that the mapping $x \in \mathcal{X} \to (c - c')^\top x$ is constant. In fact, for $x, x' \in \mathcal{X}$, we have

$$(c - c')^\top x - (c - c')^\top x' = \underbrace{(c - c')^\top}_{\in F^\perp} \underbrace{(x - x')}_{\in F_0 \cap \mathrm{Ker}\ A} = 0,$$

  by the assumption $F^\perp \perp F_0 \cap \mathrm{Ker}\ A$. Hence, the mappings $x \longmapsto c^\top x$ and $x \longmapsto c'^\top x$ are identical in $\mathcal{X}$, within a constant. Consequently, we have $\arg\min_{x \in \mathcal{X}} c^\top x = \arg\min_{x \in \mathcal{X}} c'^\top x$.

- Assume that $F^\perp \not\perp F_0 \cap \mathrm{Ker}\ A$. Let $c \in \mathbb{R}^d$. We would like to show that there exists $c' \in \mathcal{C}$ such that $c_F = c'_F$ and $\arg\min_{x \in \mathcal{X}} c^\top x \neq \arg\min_{x \in \mathcal{X}} c'^\top x$. Let $x^\star(c) \in \arg\min_{x \in \mathcal{X}} c^\top x$. There exists a set of feasible directions for $x^\star(c)$, $V = \{\delta_1, \ldots, \delta_r\} \subset \mathrm{FD}(x^\star(c))$, that spans $F_0 \cap \mathrm{Ker}\ A$ (see Lemma 7). Since $V$ spans $F_0 \cap \mathrm{Ker}\ A$, and $F^\perp \not\perp F_0 \cap \mathrm{Ker}\ A$, then there exists $\delta \in V$ such that $\mathrm{proj}_{F^\perp}(\delta) \neq 0$. Let $M$ be a positive constant and define $c' = c - M\mathrm{proj}_{F^\perp}(\delta)$. We have $c'_F = c_F$. For all $\alpha > 0$ such that $x^\star(c) + \alpha\delta \in \mathcal{X}$, we have

$$c'^\top(x^\star(c) + \alpha\delta) = c'^\top x^\star(c) + \alpha c^\top \delta - \alpha M \mathrm{proj}_{F^\perp}(\delta)^\top \delta$$
$$= c'^\top x^\star(c) + \alpha c^\top \delta - \alpha M \left\| \mathrm{proj}_{F^\perp}(\delta) \right\|^2.$$

  When $M$ is set to be large enough, we can see that we have $c'^\top(x^\star(c) + \alpha\delta) < c'^\top x^\star(c)$, which means that $x^\star(c) \notin \arg\min_{x \in \mathcal{X}} c'^\top x$.

Let us now prove the final part of the proposition. Let $K > 0$ and $\hat{x} : \mathbb{R}^N \longrightarrow \mathcal{X}$. We first prove that the set of feasible directions from $\hat{x}\left(c^\top q_1, \ldots, c^\top q_N\right)$ spans $F_0 \cap \mathrm{Ker}\ A$. We know from Lemma 7 that the set of feasible directions from any extreme point spans $F_0 \cap \mathrm{Ker}\ A$. Let $x^1, \ldots, x^\ell, \ell \in \mathbb{N}$

and $\lambda_1, \ldots, \lambda_l \in (0,1]$ such that $\hat{x}\left(c^\top q_1, \ldots, c^\top q_N\right) = \sum_{i=1}^\ell \lambda_i x_i$. For any feasible direction $\delta$ from $x^1$, for $\alpha > 0$, we have

$$\hat{x}\left(c^\top q_1, \ldots, c^\top q_N\right) + \alpha\delta = \lambda_1(x^1 + \frac{1}{\lambda_1}\alpha\delta) + \sum_{i=2}^\ell \lambda_i x^i.$$

For $\alpha$ small enough, we can see that $x^1 + \frac{1}{\lambda_1}\alpha\delta \in \mathcal{X}$ and consequently $\hat{x}\left(c^\top q_1, \ldots, c^\top q_N\right) + \alpha\delta \in \mathcal{X}$. Hence, any feasible direction from $x^1$ is feasible from $\hat{x}\left(c^\top q_1, \ldots, c^\top q_N\right)$. This means that the feasible directions from $\hat{x}\left(c^\top q_1, \ldots, c^\top q_N\right)$ span $F_0 \cap \operatorname{Ker} A$. Hence, there exist $\delta \neq 0$ a feasible direction from $\hat{x}\left(c^\top q_1, \ldots, c^\top q_N\right)$ such that $\delta_{F^\perp} \neq 0$. $c_{F^\perp}$ can take any value in $F^\perp$ without changing the values of $c^\top q_1, \ldots, c^\top q_N$. Consequently, we set $c_{F^\perp} = -M\delta_{F^\perp}$ where $M$ is a nonnegative number that we will set later. Hence, letting $\alpha > 0$ such that $\hat{x}\left(c^\top q_1, \ldots, c^\top q_N\right) + \alpha\delta \in \mathcal{X}$, we have

$$c^\top(\hat{x}\left(c^\top q_1, \ldots, c^\top q_N\right) + \alpha\delta) = c^\top \hat{x}\left(c^\top q_1, \ldots, c^\top q_N\right) + \alpha c_F^\top \delta + \alpha c_{F^\perp}^\top \delta$$
$$= c^\top \hat{x}\left(c^\top q_1, \ldots, c^\top q_N\right) + \alpha c_F^\top \delta - M\alpha \|\delta_{F^\perp}\|^2$$

This implies $c^\top \hat{x}\left(c^\top q_1, \ldots, c^\top q_N\right) + \alpha c_F^\top \delta - M\alpha \|\delta_{F^\perp}\|^2 \geq \min_{x \in \mathcal{X}} c^\top x$, i.e.

$$c^\top \hat{x}\left(c^\top q_1, \ldots, c^\top q_N\right) \geq -\alpha c_F^\top \delta + M\alpha \|\delta_{F^\perp}\|^2 + \min_{x \in \mathcal{X}} c^\top x.$$

Taking $M \geq \frac{K + \alpha c_F^\top \delta}{\alpha \|\delta_{F^\perp}\|^2}$, we indeed get

$$c^\top \hat{x}\left(c^\top q_1, \ldots, c^\top q_N\right) \geq K + \min_{x \in \mathcal{X}} c^\top x.$$

$\square$

## A.4 Proof of Proposition 5

*Proof.* Let $x^\star \in \mathcal{X}^{\angle}$. We denote $J = \{i \in [d], x_i^\star = 0\}$ and $I_0 = \{i \in [d], \exists x \in \mathcal{X}, x_i \neq 0\}$. For every $\delta \in \mathbb{R}^d$, we have

$$\delta \in \operatorname{FD}(x^\star) \iff \exists \varepsilon > 0, \ x^\star + \varepsilon\delta \geq 0 \text{ and } A\delta = 0 \iff A\delta = 0 \text{ and } \delta_j \geq 0 \text{ for every } j \in J.$$

This means that $\operatorname{FD}(x^\star)$ is a polyhedral cone, and $\operatorname{FD}(x^\star) \subset \operatorname{Ker} A$. Furthermore, since $[d] \setminus I_0 \subset J$, we also have $\operatorname{FD}(x^\star) \subset F_0$ which yields $\operatorname{FD}(x^\star) \subset F_0 \cap \operatorname{Ker} A$. $\square$

## A.5 Proof of Proposition 6

*Proof.* Let $x^\star \in \mathcal{X}^{\angle}$. For every $c \in \mathbb{R}^d$, we have

$$x^\star \in \arg\min_{x \in \mathcal{X}} c^\top x \iff \forall \delta \in \operatorname{FD}(x^\star), \ c^\top \delta \geq 0 \iff \forall \delta \in D(x^\star), \ c^\top \delta \geq 0.$$

$\square$

## A.6 Proof of Theorem 1

*Proof.* We denote $F = \operatorname{span} \mathcal{D}$. Notice that we have $\Delta(\mathcal{X}, \mathcal{C}) \subset F \iff \Delta(\mathcal{X}, \mathcal{C}) \perp F^\perp$. We will now prove that $\mathcal{D}$ is a sufficient decision dataset for $\mathcal{C}$ if and only if $\Delta(\mathcal{X}, \mathcal{C}) \perp F^\perp$.

- ($\Leftarrow$) Suppose $\Delta(\mathcal{X}, \mathcal{C}) \not\perp \cap F^\perp$. There exists $\delta \in \Delta(\mathcal{X}, \mathcal{C})$ such that $\delta \not\perp F^\perp$. By definition, there exists $x \in \mathcal{X}^{\angle}$ such that $\delta \in D(x)$ and $F(x, \delta) \cap \mathcal{C} \neq \varnothing$. Let $v \in F(x, \delta) \cap \mathcal{C}$. Let $\delta_0 \in F^\perp$ such that $\delta_0^\top \delta < 0$ ($\delta_0$ exists because $\delta \not\perp F^\perp$). As $\mathcal{C}$ is open, we can assume without loss of generality that $v + \delta_0 \in \mathcal{C}$ by rescaling $\delta_0$. We know that $v \in F(x, \delta) \subset \Lambda(x)$, and $(v + \delta_0)^\top \delta = \delta_0^\top \delta < 0$ which implies that $v + \delta_0 \notin \Lambda(x)$. Finally, since we have $\delta_0 \in F^\perp$, we have $(v + \delta_0)_F = v_F + \delta_{0,F} = v_F$. However, $v \in \Lambda(x)$ and $v + \delta \notin \Lambda(x)$ which implies $x \in \arg\min_{x' \in \mathcal{X}} v^\top x'$ and $x \notin \arg\min_{x' \in \mathcal{X}} (v + \delta_0)^\top x'$, meaning that $\arg\min_{x' \in \mathcal{X}} (v + \delta_0)^\top x' \neq \arg\min_{x' \in \mathcal{X}} v^\top x'$. This implies that $\mathcal{D}$ is not a sufficient from Proposition 2.

- ($\Rightarrow$) Suppose $\mathcal{D}$ is not sufficient. From Proposition 2, there exists $c, c' \in \mathcal{C}$ such that $c_F = c'_F$ and $\arg\min_{x \in \mathcal{X}} c^\top x \neq \arg\min_{x \in \mathcal{X}} c'^\top x$. It follows from the definition of the optimality cones $\Lambda$ (Proposition 6) that there exists $x \in \mathcal{X}^\angle$ such that $c \in \Lambda(x)$ and $c' \notin \Lambda(x)$ (see also Lemma 6). For any $\alpha \in [0, 1]$, we denote $c_\alpha := (1 - \alpha)c + \alpha c'$

$$\alpha^\star := \sup\{\alpha \in [0, 1] \ : \ c_\alpha \in \Lambda(x)\}$$
$$= \sup\{\alpha \in [0, 1] \ : \ c_\alpha^\top \delta \geq 0, \ \forall \delta \in D(x)\}.$$

Since $\mathcal{C}$ is convex, we have $c_{\alpha^\star} \in \mathcal{C}$. Since $\Lambda(x)$ is a closed set, we have $c_{\alpha^\star} \in \Lambda(x)$ and hence we have $c_{\alpha^\star} \neq c'$ i.e. $\alpha^\star < 1$. Let $\varepsilon \in (0, 1 - \alpha^\star)$ small enough such that for any $\delta \in D(x)$ such that $c_{\alpha^\star}{}^\top \delta > 0$, we have $c_{\alpha^\star + \varepsilon}^\top \delta > 0$. As $c_{\alpha^\star + \epsilon} \notin \Lambda(x)$, there exists $\delta \in D(x)$ for which $c_{\alpha^\star + \epsilon}{}^\top \delta < 0$. Such $\delta$ must verify $c_{\alpha^\star}{}^\top \delta = 0$ given the condition defining $\epsilon$. Hence, $c_{\alpha^\star} \in \Lambda(x) \cap \{\delta\}^\perp = F(x, \delta)$, and $c_{\alpha^\star} \in \mathcal{C}$, which implies $F(x, \delta) \cap \mathcal{C} \neq \varnothing$ and therefore $\delta \in \Delta(\mathcal{X}, \mathcal{C})$. Moreover, we have $\underbrace{(c_{\alpha^\star + \varepsilon} - c_{\alpha^\star})}_{=\varepsilon(c'-c) \in F^\perp}{}^\top \delta = c_{\alpha^\star + \varepsilon}^\top \delta \neq 0$, i.e. $\delta \not\perp F^\perp$,

and consequently we have $\Delta(\mathcal{X}, \mathcal{C}) \not\perp F^\perp$.

$\square$

## A.7 Proof of Theorem 2

Before proving the theorem, we will have to introduce a few lemmas and definition.

**Lemma 2.** *For any $c \in \mathbb{R}^d$, there exists $\varepsilon > 0$ such that for any $c'$ satisfying $\|c - c'\| < \varepsilon$, $\arg\min_{x \in \mathcal{X}} c^\top x \cap \arg\min_{x \in \mathcal{X}} c'^\top x \neq \varnothing$.*

*Proof.* Assume that there exists $c \in \mathbb{R}^d$ such that for all $\varepsilon > 0$, there exists $c'$ satisfying $\|c - c'\| < \varepsilon$ and $\arg\min_{x \in \mathcal{X}} c^\top x \cap \arg\min_{x \in \mathcal{X}} c'^\top x = \varnothing$. There exists a sequence $(c'_n)_{n \in \mathbb{N}}$ that converges to $c$ such that for all $n \in \mathbb{N}$, there exists $x \in \mathcal{X}^\angle \setminus \arg\min_{x \in \mathcal{X}} c^\top x$ such that $x \in \arg\min_{x \in \mathcal{X}} c_n'^\top x$. Since there is a finite number of extreme points, there exists a subsequence $(c'_{\varphi(n)})_{n \in \mathbb{N}}$ and $x \in \mathcal{X}^\angle \setminus \arg\min_{y \in \mathcal{X}} c^\top y$ such that for all $n \in \mathbb{N}$, we have $x \in \arg\min_{y \in \mathcal{X}} c_{\varphi(n)}'^\top y$, i.e. $c'_{\varphi(n)} \in \Lambda(x)$. Hence, since $\Lambda(x)$ is closed, we have $c \in \Lambda(x)$ and $x \notin \arg\min_{y \in \mathcal{X}} c^\top y$ which is not possible. $\square$

**Definition 5** (Extreme Point Neighbors). Let $\mathcal{C} \subset \mathbb{R}^d$. For any two extreme points $x_1, x_2 \in X^\angle$, we say that $x_1$ and $x_2$ are neighbors in $\mathcal{X}$ if there exists an extreme direction $\delta \in D(x_1)$ such that $x_2 = x_1 + \delta$. We say that they are $\mathcal{C}-$strong neighbors in $\mathcal{X}$ if furthermore there exists $c \in \mathcal{C}$ such that $x, x' \in \arg\min_{y \in \mathcal{X}} c^\top y$.

**Definition 6** (Connected and $\mathcal{C}-$strongly Connected Points). For any subset $\mathcal{Y} \subset \mathcal{X}^\angle$ and $\mathcal{C} \subset \mathbb{R}^d$, for any pair of elements $x, x' \in \mathcal{Y}$, we say that $x, x'$ are connected by neighboring extreme points in $\mathcal{Y}$ if there exist $h \in \mathbb{N}$ and a sequence $x_1, \ldots, x_h \in \mathcal{Y}$ such that for all $i \in [h-1]$, $x_i$ and $x_{i+1}$ are neighbors in $\mathcal{X}$ and $x_1 = x$ and $x_h = x'$. We say that they are $\mathcal{C}-$strongly connected, when $x_i$ and $x_{i+1}$ are $\mathcal{C}-$strong neighbors. When there is no ambiguity, we say that $x$ and $x'$ are (strongly) connected.

We say that the set $\mathcal{Y}$ is ($\mathcal{C}-$strongly) connected by neighboring extreme points if this property holds for any pair of extreme points in $\mathcal{Y}$. When there is no ambiguity, we say that $\mathcal{Y}$ is (strongly) connected. For any element $x$ of $\mathcal{Y}$, we call the ($\mathcal{C}-$strong) connection class of $x$ the set of points in $\mathcal{Y}$ that are ($\mathcal{C}-$strongly) connected by neighboring extreme points to $x$.

**Lemma 3.** *For any $c \in \mathcal{C}$, $\mathcal{X}^\angle \cap \arg\min_{x \in \mathcal{X}} c^\top x$ is $\mathcal{C}-$strongly connected by neighboring extreme points in $\mathcal{X}$.*

*Proof.* Let $c \in \mathcal{C}$. Every extreme point in $\arg\min_{x \in \mathcal{X}} c^\top x$ is also an extreme point in $\mathcal{X}$ (see Lemma 6), and every extreme direction in $\arg\min_{x \in \mathcal{X}} c^\top x$ is also an extreme direction in $\mathcal{X}$. Hence, since $\arg\min_{x \in \mathcal{X}} c^\top x$ is a bounded polyhedron, $\mathcal{X}^\angle \cap \arg\min_{x \in \mathcal{X}} c^\top x$ is connected by neighboring extreme points in $\mathcal{X}$. Furthermore, by definition, since $\mathcal{X}^\angle \cap \arg\min_{x \in \mathcal{X}} c^\top x \subset \arg\min_{x \in \mathcal{X}} c^\top x$, then $\mathcal{X}^\angle \cap \arg\min_{x \in \mathcal{X}} c^\top x$ is $\mathcal{C}-$strongly connected. $\square$

**Lemma 4.** *When $\mathcal{C}$ is convex, $\mathcal{X}^\star(\mathcal{C}) \cap \mathcal{X}^{\angle}$ is $\mathcal{C}-$strongly connected by neighboring extreme points.*

*Proof.* Assume that there exist $x, x' \in \mathcal{X}^\star(\mathcal{C}) \cap \mathcal{X}^{\angle}$ that are not strongly connected. Let $c, c' \in \mathcal{C}$ such that $x \in \arg\min_{y \in \mathcal{X}} c^\top y$, $x' \in \arg\min_{y \in \mathcal{X}} c'^\top y$. For any $\alpha \in [0,1]$, we denote $c_\alpha := (1-\alpha)c + \alpha c'$. Let

$$U := \{x^\star \in \mathcal{X}^{\angle}, \ \exists \alpha \in [0,1], \ x^\star \in \arg\min_{y \in \mathcal{X}} c_\alpha^\top y\} \subset \mathcal{X}^\star(\mathcal{C}) \cap \mathcal{X}^{\angle}.$$

Let $K$ be the intersection of $U$ and the connection class of $x$. We have $x' \notin K$. Let

$$\alpha^\star = \max\left\{\alpha \in [0,1], \ K \cap \arg\min_{y \in \mathcal{X}} c_\alpha^\top y \neq \varnothing\right\}.$$

If $\alpha^\star = 1$, then there exists $v \in \arg\min_{y \in \mathcal{X}} c'^\top y$ such that $v \in K$. From Lemma 3, $\mathcal{X}^{\angle} \cap \arg\min_{y \in \mathcal{X}} c'^\top y$ is $\mathcal{C}-$strongly connected and $v \in K \cap \mathcal{X}^{\angle} \cap \arg\min_{y \in \mathcal{X}} c'^\top y$ and consequently $x' \in K$, and therefore is connected to $x$ which contradicts our assumption. Hence, we necessarily have $\alpha^\star < 1$. Furthermore, from Lemma 2, there exists $\varepsilon \in (0, 1 - \alpha^\star)$ such that

$$\arg\min_{y \in \mathcal{X}} c_{\alpha^\star + \varepsilon}^\top y \cap \arg\min_{y \in \mathcal{X}} c_{\alpha^\star}^\top y \neq \varnothing. \tag{3}$$

As $K \cap \arg\min_{y \in \mathcal{X}} c_{\alpha^\star}^\top y \neq \varnothing$ and $\arg\min_{y \in \mathcal{X}} c_{\alpha^\star}^\top y$ is $\mathcal{C}-$strongly connected from Lemma 3, we have $\arg\min_{y \in \mathcal{X}} c_{\alpha^\star}^\top y \subset K$. Combined with (3), it implies that $K \cap \arg\min_{y \in \mathcal{X}} c_{\alpha^\star + \varepsilon}^\top y \neq \varnothing$. This contradicts the supremum definition of $\alpha^\star$. $\square$

We have now enough tools to prove the theorem.

*Proof of Theorem 2.* We have

$$\mathrm{span}\,\Delta(\mathcal{X}, \mathcal{C}) \underset{(1)}{=} \mathrm{span}\,\{x_1 - x_2, \ x_1, x_2 \in \mathcal{X}^{\angle} \cap \mathcal{X}^\star(\mathcal{C}), \ x_1 \text{ and } x_2 \text{ are } \mathcal{C}-\text{strong neighbors}\}$$
$$\tag{4}$$

$$\underset{(2)}{=} \mathrm{span}\,\{x_1 - x_2, \ x_1, x_2 \in \mathcal{X}^{\angle} \cap \mathcal{X}^\star(\mathcal{C})\} \tag{5}$$

$$\underset{(3)}{=} \mathrm{dir}\left(\mathcal{X}^{\angle} \cap \mathcal{X}^\star(\mathcal{C})\right) \tag{6}$$

$$\underset{(4)}{=} \mathrm{dir}\left(\mathcal{X}^\star(\mathcal{C})\right). \tag{7}$$

Let's justify each of the equalities above.

- (1) Let $\delta \in \Delta(\mathcal{X}, \mathcal{C})$. There exists $c \in \mathcal{C}$ and $x \in \mathcal{X}^{\angle}$ such that $c \in F(x, \delta)$. This means that $x \in \arg\min_{y \in \mathcal{X}} c^\top y$, $\delta$ is an extreme direction for $x$ in $\mathcal{X}$, and $c^\top \delta = 0$. Consequently, there exists $\eta > 0$ such that $x' := x + \eta\delta$ is an extreme point, that is a neighbor of $x$ by definition. Also, we have $x' \in \arg\min_{y \in \mathcal{X}} c^\top y$. Hence, $\delta = \frac{1}{\eta}(x' - x)$, which proves

  $$\Delta(\mathcal{X}, \mathcal{C}) \subset \mathrm{span}\,\{x_1 - x_2, \ x_1, x_2 \in \mathcal{X}^{\angle} \cap \mathcal{X}^\star(\mathcal{C}), \ x_1 \text{ and } x_2 \text{ are } \mathcal{C}-\text{strong neighbors}\}.$$

  Conversely, if $x_1, x_2 \in \mathcal{X}^{\angle} \cap \mathcal{X}^\star(\mathcal{C})$ are $\mathcal{C}-$strong neighbors, then there exists an extreme direction $\delta$ for $x_1$ such that $x_2 = x_1 + \delta$ and $c \in \mathcal{C}$ such that $x_1, x_2 \in \arg\min_{y \in \mathcal{X}} c^\top y$. Hence, we have $c^\top \delta = 0$, and consequently $c \in F(x_1, \delta)$, which means that $\delta \in \Delta(\mathcal{X}, \mathcal{C})$, i.e. $x_1 - x_2 \in \Delta(\mathcal{X}, \mathcal{C})$. This proves the desired equality.

- (2) Set Eq. (4) is clearly a subset of set Eq. (5). Let's prove the converse inclusion. Let $x, x' \in \mathcal{X}^{\angle} \cap \mathcal{X}^\star(\mathcal{C})$. According to Lemma 4, there exists $h \in \mathbb{N}$ and a sequence $x_1, \ldots, x_h \in \mathcal{X}^{\angle} \cap \mathcal{X}^\star(\mathcal{C})$ such that $x_1 = x$ and $x_h = x'$ and for all $i \in [h-1]$, $x_i, x_{i+1}$ are $\mathcal{C}-$strongly connected. Hence, we have

  $$x - x' = \sum_{i=1}^{h-1} x_{i+1} - x_i.$$

All of the terms in the sum above are in set (4) and therefore their sum as well, by linearity. Hence, we indeed have the inclusion.

- (3) This equality is immediate since for any $x_1, x_2 \in \mathcal{X}^{\measuredangle} \cap \mathcal{X}^\star(\mathcal{C})$, $x_1 - x_2 = x_1 - x_0 - (x_2 - x_0)$ for any $x_0 \in \mathcal{X}^{\measuredangle} \cap \mathcal{X}^\star(\mathcal{C})$ and consequently $x_1 - x_2 \in \text{dir}\left(\mathcal{X}^{\measuredangle} \cap \mathcal{X}^\star(\mathcal{C})\right)$.

- (4) In order to prove this equality, we prove that $\mathcal{X}^\star(\mathcal{C}) \subset \text{conv}(\mathcal{X}^{\measuredangle} \cap \mathcal{X}^\star(\mathcal{C}))$. Let $x \in \mathcal{X}^\star(\mathcal{C})$ and $c \in \mathcal{C}$ such that $x \in \arg\min_{y \in \mathcal{X}} c^\top y$. There exists $\alpha_1, \ldots, \alpha_k \in (0, 1]$ such that $\alpha_1 + \cdots + \alpha_k = 1$ and $x_1, \ldots, x_k \in \mathcal{X}^{\measuredangle}$ such that $x = \sum_{i=1}^k \alpha_k x_k$. We have

$$\min_{y \in \mathcal{X}} c^\top y \geq \sum_{i=1}^k \alpha_k c^\top x_k \text{ i.e. } \sum_{i=1}^k \alpha_k (c^\top x_k - \min_{y \in \mathcal{X}} c^\top y) \leq 0.$$

All of the terms in the sum are positive, and are consequently equal to 0. Hence we have $x_1, \ldots, x_k \in \arg\min_{y \in \mathcal{X}} c^\top y \subset \mathcal{X}^\star(\mathcal{C})$. Consequently, we have $x \in \text{conv}(\mathcal{X}^{\measuredangle} \cap \mathcal{X}^\star(\mathcal{C}))$. Hence, we have

$$\text{dir}\left(\mathcal{X}^\star(\mathcal{C})\right) \subset \text{dir}\left(\text{conv}(\mathcal{X}^{\measuredangle} \cap \mathcal{X}^\star(\mathcal{C}))\right)$$
$$= \text{dir}\left(X^{\measuredangle} \cap \mathcal{X}^\star(\mathcal{C})\right)$$
$$\subset \text{dir}\left(\mathcal{X}^\star(\mathcal{C})\right).$$

This proves the desired equality.

$\square$

## A.8 Proof of Proposition 1

Before proving the proposition, we need to introduce the following lemma.

**Lemma 5.** *For any $x^\star \in \mathcal{X}^{\measuredangle}$, there exists $c \in \mathbb{R}^d$ such that $\arg\min_{x \in \mathcal{X}} c^\top x = \{x^\star\}$, i.e. for all $\delta \in D(x^\star)$, $c^\top \delta > 0$.*

*Proof.* Let $x^\star \in \mathcal{X}^{\measuredangle}$. Assume that such a $c \in \mathbb{R}^d$ does not exists. We first show that there exists $\delta^\star \in D(x^\star)$ such that $\Lambda(x^\star) \perp \delta^\star$. Suppose no such $\delta^\star$ exists, then for any $\delta \in D(x^\star)$, there would exist $v(\delta) \in \Lambda(x^\star)$ such that $v(\delta)^\top \delta > 0$. Consequently, we have for any $\delta \in D(x^\star)$,

$$\left(\sum_{\delta' \in D(x^\star)} v(\delta')\right)^\top \delta > 0,$$

which contradicts our initial assumption.

Let $N \in \mathbb{N}$ and $\delta_1, \ldots, \delta_N$ such that $D(x^\star) = \{\delta_1, \ldots, \delta_N\}$. Assume without loss of generality that $\Lambda(x^\star) \perp \delta_N$. Consequently, we have for all $c \in \mathbb{R}^d$

$$\left(\forall i \in [N-1], \; c^\top \delta_i \geq 0\right) \implies c^\top \delta_N \leq 0.$$

We show that this implies that $-\delta_N$ belongs to the cone spanned by $\delta_1, \ldots, \delta_{N-1}$, i.e. there exists $\mu_1, \ldots, \mu_{N-1} \in \mathbb{R}^+$ such that $\sum_{i=1}^{N-1} \mu_i \delta_i = -\delta_N$. Assume that this is not true. Let $K$ be the cone spanned by $\delta_1, \ldots, \delta_{N-1}$. Since $-\delta_N \notin K$, then (by the separation lemma), there exists $u \in \mathbb{R}^d$ such that for all $h \in K$, we have $u^\top h \geq 0$ and $-u^\top \delta_N < 0$. In particular, we have for all $i \in [N-1]$, $u^\top \delta_i \geq 0$ and $u^\top \delta_N > 0$, a contradiction. Hence, there exists $\alpha_1, \ldots, \alpha_{N-1} \in \mathbb{R}^+$ such that

$$-\delta_N = \sum_{i=1}^{N-1} \alpha_i \delta_i.$$

Consequently, both $\delta_N$ and $-\delta_N$ are feasible directions from $x^\star$ in $\mathcal{X}$, which contradicts the fact that $x^\star$ is an extreme point. $\square$

We now prove Proposition 1.

*Proof.* It is easy to see that 1 implies 2. We now prove that 2 implies 1. Assume that 2 is verified but not 1, that is $\mathcal{D}$ is not a sufficient decision dataset. From Theorem 1, there exists $\delta \in \Delta(\mathcal{X}, \mathcal{C})$ such that $\delta \not\perp (\text{span } \mathcal{D})^\perp$. By definition, there exists $x^\star \in \mathcal{X}^\angle$ and $c \in \mathcal{C}$ such that $c \in F(x^\star, \delta)$. From Lemma 5, there exists $v \in \mathbb{R}^d$ such that for all $\delta \in D(x^\star)$, $v^\top \delta > 0$. Let $\varepsilon > 0$ such that $B(c, \varepsilon) \subset \mathcal{C}$. Let $\eta > 0$ small enough such that $c + \eta v \in B(c, \varepsilon)$, and $\eta'$ small enough such that $c_{\eta, \eta'} := c + \eta v - \eta' \delta_{(\text{span } \mathcal{D})^\perp} \in B(c, \varepsilon)$. For any $\delta' \in D(x^\star)$, we have

$$(c + \eta v)^\top \delta' = \underbrace{c^\top \delta'}_{\geq 0} + \underbrace{\eta v^\top \delta'}_{>0} > 0.$$

This means that $\arg\min_{x \in \mathcal{X}} (c + \eta v)^\top x = \{x^\star\}$. Furthermore, we have

$$c_{\eta, \eta'}^\top \delta = \underbrace{c^\top \delta}_{=0, \text{ as } c \in F(x^\star, \delta)} + \eta v^\top \delta - \eta' \underbrace{\left\| \delta_{(\text{span } (\mathcal{D}))^\perp} \right\|^2}_{\neq 0, \text{ as } \delta \not\perp (\text{span } \mathcal{D})^\perp}.$$

Consequently, when $\eta$ is small enough compared to $\eta'$, we have $c_{\eta, \eta'}^\top \delta < 0$, i.e. $c_{\eta, \eta'} \notin \Lambda(x^\star)$. This means that $x^\star \notin \arg\min_{x \in \mathcal{X}} c_{\eta, \eta'}^\top x$. Assume that a mapping $\hat{x}$ satisfying condition 2 of the proposition. We have $c + \eta v - c_{\eta, \eta'} = \eta' \delta_{(\text{span } (\mathcal{D}))^\perp} \in (\text{span } (\mathcal{D}))^\perp$. This means that for all $i \in [N]$, we have $(c + \eta v)^\top q_i = c_{\eta, \eta'}^\top q_i$, and hence

$$\hat{x}((c + \eta v)^\top q_1, \ldots, (c + \eta v)^\top q_N) = \hat{x}(c_{\eta, \eta'}^\top q_1, \ldots, c_{\eta, \eta'}^\top q_N),$$

which implies that

$$\hat{x}(c_{\eta, \eta'}^\top q_1, \ldots, c_{\eta, \eta'}^\top q_N) \in \left( \arg\min_{x \in \mathcal{X}} c_{\eta, \eta'}^\top x \right) \cap \left( \arg\min_{\mathcal{X}} (c + \eta v)^\top x \right) = \varnothing,$$

which is impossible.

$\square$

# B  Detailed Algorithm and Correction Proof

## B.1  Algorithm to find a basis of $\dim (\mathcal{X}^\star (\mathcal{C}))$

---
**Algorithm 2** Computing $\dim (\mathcal{X}^\star (\mathcal{C}))$

---
**Input:** Polyhedron $\mathcal{X} = \{x \geq 0 \; : \; Ax = b\}$, Uncertainty set $\mathcal{C}$.
**Output:** A basis of $\dim (\mathcal{X}^\star (\mathcal{C}))$.
Initialize $\mathcal{D}$ to $\varnothing$.
Set $x_0 \in \arg\min_{x \in \mathcal{X}} c_0^\top x$ for some $c_0 \in \mathcal{C}$.
Sample $\alpha \sim \mathcal{N}(0, Id)$.
**while** either of the problems

$$\min / \max \alpha^\top \text{proj}_{(\text{span } \mathcal{D})^\perp} (x_0 - x)$$
$$\text{s.t. } x \geq 0, \; \lambda \in \mathbb{R}^m, \; s \in \mathbb{R}_+^d, \; c \in \mathcal{C}$$
$$Ax = b, \; A^\top \lambda + s = c,$$
$$1 - \epsilon s_i \geq \tau_i \geq \epsilon x_i, \; \tau_i \in \{0, 1\}, \; \forall i$$

has a solution $x^\star$ with non-zero optimal value,
$\mathcal{D} \leftarrow \mathcal{D} \cup \{x^\star - x_0\}$.
resample $\alpha \sim \mathcal{N}(0, Id)$.
**return** $\mathcal{D}$

---

## B.2  Full algorithm: Data Selection and Induced Decision

Here, $e_1, \ldots, e_d$ is the canonical basis of $\mathbb{R}^d$.

---

**Algorithm 3** Data Selection Under Query Constraints

---

**Input:** Polyhedron $\mathcal{X}$, Uncertainty Set $\mathcal{C}$, Query Set $\mathcal{Q} = \{q_1, \ldots, q_d\}$ (basis of $\mathbb{R}^d$)
**Output:** A minimal sufficient dataset under constraint $\mathcal{D} \subset \mathcal{Q}$.
Find $\{v_1, \ldots, v_k\}$ a basis of $\dim(\mathcal{X}^\star(\mathcal{C}))$ using Algorithm 2
$Q \leftarrow [q_1, \ldots, q_d]$
**return** $\mathcal{D} := \{q_i \ : \ i \in [d] \text{ s.t. } \exists j \in [k], \ (Q^{-1}v_j)^\top e_i \neq 0\}$.

---

---

**Algorithm 4** Decision-making with a Sufficient Decision Dataset

---

**Input:** Decision set $\mathcal{X}$, Uncertainty Set $\mathcal{C}$, Sufficient Decision Dataset $\mathcal{D} = \{q_1, \ldots, q_N\}$, Oracle $\pi$
      such that for any $q \in \mathcal{Q}$, $\pi(q) = c^\top q$ where $c$ is the ground truth.
**Output:** A decision $\hat{x} \in \arg\min_{x \in \mathcal{X}} c^\top x$.
$o_1, \ldots, o_N \leftarrow \pi(q_1), \ldots, \pi(q_N)$
Compute $\hat{c} \in \arg\min\{\sum_{i=1}^N (c'^\top q_i - o_i)^2 \ : \ c' \in \mathcal{C}\}$.
**return** $\hat{x} \in \arg\min_{x \in \mathcal{X}} \hat{c}^\top x$.

---

## B.3  Proof of Theorem 3: Correctness

*Proof.*

- We first show that when the algorithm terminates, i.e., the condition of the while loop is no longer satisfied, then with probability 1, $\dim(\mathcal{X}^\star(\mathcal{C})) \subset \operatorname{span}\mathcal{D}$. Notice that the constraints in the minimization and maximization problems in Algorithm 2 encode complimentary slackness and, therefore are equivalent to

$$\min/\max\{\alpha^\top \operatorname{proj}_{(\operatorname{span}\mathcal{D})^\perp}(x^\star - x_0) \ : \ c \in \mathcal{C}, \ x^\star \in \arg\min_{x \in \mathcal{X}} c^\top x\}.$$

  By definition of $\mathcal{X}^\star(\mathcal{C})$, this equivalent to

$$\min/\max\{\alpha^\top \operatorname{proj}_{(\operatorname{span}\mathcal{D})^\perp}(x^\star - x_0) \ : \ x^\star \in \mathcal{X}^\star(\mathcal{C})\}.$$

  If the two problems have an optimal value equal to 0, then $\operatorname{proj}_{(\operatorname{span}\mathcal{D})^\perp}(\dim(\mathcal{X}^\star(\mathcal{C}))) \perp \alpha$ i.e. $\alpha \in \operatorname{proj}_{(\operatorname{span}\mathcal{D})^\perp}(\dim(\mathcal{X}^\star(\mathcal{C})))^\perp$. Unless $\operatorname{proj}_{(\operatorname{span}\mathcal{D})^\perp}(\dim(\mathcal{X}^\star(\mathcal{C})))^\perp = \mathbb{R}^d$, this set is of empty interior and its Lebesgue measure is equal to 0, and consequently the probability of having $\operatorname{proj}_{(\operatorname{span}\mathcal{D})^\perp}(\dim(\mathcal{X}^\star(\mathcal{C}))) \perp \alpha$ is zero since $\alpha$ has a continuous distribution. Hence, with probability 1, we have $\operatorname{proj}_{(\operatorname{span}\mathcal{D})^\perp}(\dim(\mathcal{X}^\star(\mathcal{C}))) = \{0\}$ i.e. $\dim(\mathcal{X}^\star(\mathcal{C})) \subset \operatorname{span}\mathcal{D}$.

- We now show that at every step of the algorithm, the dimension of the span of $\mathcal{D}$ increases by 1, and that it remains a linearly independent set, as well as satisfies $\operatorname{span}\mathcal{D} \subset \dim(\mathcal{X}^\star(\mathcal{C}))$. Indeed, initially, $\mathcal{D}$ is empty and is hence a linearly independent set and satisfies $\operatorname{span}\mathcal{D} \subset \dim(\mathcal{X}^\star(\mathcal{C}))$. Assuming that $\mathcal{D}$ is a linearly independent set and that $\operatorname{span}\mathcal{D} \subset \dim(\mathcal{X}^\star(\mathcal{C}))$, if there exists $x \in \mathcal{X}^\star(\mathcal{C})$ such that $\alpha^\top \operatorname{proj}_{(\operatorname{span}\mathcal{D})^\perp}(x_0 - x) \neq 0$, then $\operatorname{proj}_{(\operatorname{span}\mathcal{D})^\perp}(x_0 - x) \neq 0$ with probability 1 and consequently $x_0 - x \in \dim(\mathcal{X}^\star(\mathcal{C})) \backslash \operatorname{span}\mathcal{D}$. Hence, we have $\dim(\operatorname{span}(\mathcal{D} \cup \{x_0 - x\})) = \dim(\operatorname{span}\mathcal{D}) + 1$ and $\mathcal{D} \cup \{x_0 - x\}$ is a linearly independent set and satisfies $\operatorname{span}(\mathcal{D} \cup \{x_0 - x\}) \subset \dim(\mathcal{X}^\star(\mathcal{C}))$, which proves the desired result.

- Finally, combining the two results above, when the algorithm terminates, $\mathcal{D}$ is a linearly independent set, and $\operatorname{span}\mathcal{D} = \dim(\mathcal{X}^\star(\mathcal{C}))$ i.e. $\mathcal{D}$ is a basis of $\dim(\mathcal{X}^\star(\mathcal{C}))$ with probability 1. Furthermore, the analysis above show that the algorithm indeed terminates after $\dim\dim(\mathcal{X}^\star(\mathcal{C}))$ iterations of the while loop.

$\square$

## C Useful Lemmas

**Lemma 6.** *Assume that $\mathcal{X}$ is bounded. For every $c \in \mathbb{R}^d$, $\arg\min_{x \in \mathcal{X}} c^\top x$ is a polyhedron, and all of its extreme points are extreme points in $\mathcal{X}$. Recall the optimality cones $\Lambda(x^\star)$ of all $x^\star \in \mathcal{X}^{\angle}$ defined in Proposition 6. For every $c, c' \in \mathbb{R}^d$, the following equivalence holds.*

$$\arg\min_{x \in \mathcal{X}} c^\top x = \arg\min_{x \in \mathcal{X}} c'^\top x \iff \forall x \in \mathcal{X}^{\angle}, \; (c \in \Lambda(x) \iff c' \in \Lambda(x)).$$

*Proof.* We first show that for any $c \in \mathbb{R}^d$, any extreme point in $\arg\min_{x \in \mathcal{X}} c^\top x$ is in $\mathcal{X}^{\angle}$. Let $x \in \mathcal{X}$ be an extreme point of $\arg\min_{x \in \mathcal{X}} c^\top x$. Assume that $x$ is not an extreme point in $\mathcal{X}$. Hence, there exists $u \in \mathbb{R}^d$ such that $x \pm u \in \mathcal{X}$. If $c^\top u \neq 0$ we have $c^\top(x - u) < c^\top x$ or $c^\top(x + u) < c^\top x$. This means that $x \notin \arg\min_{x \in \mathcal{X}} c^\top x$ which is impossible. If $c^\top u = 0$, then $c \pm u \in \arg\min_{x \in \mathcal{X}} c^\top x$, which is also impossible since $x$ is an extreme point in $\arg\min_{x \in \mathcal{X}} c^\top x$. Hence, since $\arg\min_{x \in \mathcal{X}} c^\top x$ is convex, it's the convex hull of its extreme points.

Consequently, for any $c, c' \in \mathbb{R}^d$, $\arg\min_{x \in \mathcal{X}} c^\top x = \arg\min_{x \in \mathcal{X}} c'^\top x$ if and only if these two sets have the same set of extreme points. Furthermore, for any $x \in \mathcal{X}^{\angle}$, we have $c \in \Lambda(x)$ if on and only if $x \in \arg\min_{x \in \mathcal{X}} c^\top x$. Hence, the desired result immediately follows:

$$\arg\min_{x \in \mathcal{X}} c^\top x = \arg\min_{x \in \mathcal{X}} c'^\top x \iff \left( \forall x \in \mathcal{X}^{\angle}, \; x \in \arg\min_{x \in \mathcal{X}} c^\top x \iff x \in \arg\min_{x \in \mathcal{X}} c'^\top x \right)$$

$$\iff \left( \forall x \in \mathcal{X}^{\angle}, \; c \in \Lambda(x) \iff c' \in \Lambda(x) \right)$$

$\square$

**Lemma 7.** *Let $r = \dim F_0 \cap \operatorname{Ker} A$. For any $x \in \mathcal{X}$, there exists a set $V = \{\delta_1, \ldots, \delta_r\} \subset \operatorname{FD}(x)$, such that $V$ is a basis of $F_0 \cap \operatorname{Ker} A$. In particular, the set of extreme directions of $\operatorname{FD}(x)$ spans $F_0 \cap \operatorname{Ker} A$.*

*Proof.* Let $x \in \mathcal{X}$. Let $\mathcal{X}^{\angle}$ be the set of extreme points of $\mathcal{X}$, and $D^{\angle}$ be the set of extreme rays of $\mathcal{X}$. For every $x^{\angle} \in \mathcal{X}^{\angle}$ and $\delta^{\angle} \in D^{\angle}$, we have $x^{\angle} \geq 0$ and $\delta^{\angle} \geq 0$. Let $\{\alpha_{x^{\angle}}\}_{x^{\angle} \in \mathcal{X}^{\angle}} \subset \mathbb{R}_+^*$ and $\{\alpha_{\delta^{\angle}}\}_{\delta^{\angle} \in D^{\angle}} \subset \mathbb{R}_+^*$ be a set of strictly positive numbers such that $\sum_{x^{\angle} \in \mathcal{X}^{\angle}} \alpha_{x^{\angle}} = 1$. We define

$$\overline{x} = \sum_{x^{\angle} \in \mathcal{X}^{\angle}} \alpha_{x^{\angle}} x^{\angle} + \sum_{\delta^{\angle} \in D^{\angle}} \alpha_{\delta^{\angle}} \delta^{\angle} \in \mathcal{X}.$$

We have for any $i \in I_0$, $\overline{x}_i > 0$. Indeed, for any $i \in I_0$, if $\overline{x}_i = 0$, then for every $x^{\angle} \in \mathcal{X}^{\angle}$ and $\delta^{\angle} \in D^{\angle}$, $x_i^{\angle} = \delta_i^{\angle} = 0$ and hence for any $x' \in \mathcal{X}$, $x_i' = 0$ i.e. $i \notin I_0$ which is impossible. Let

$$\varepsilon := \frac{1}{2} \min_{i \in I_0} \overline{x}_i > 0,$$

for any $\delta \in F_0 \cap \operatorname{Ker} A$ such that $\|\delta\| < \varepsilon$, we have $A(\overline{x} + \delta) = A\overline{x} = b$, and for every $i \in [d]$, if $i \in I_0$, then $\overline{x}_i + \delta_i > 0$ and if $i \notin I_0$, $\overline{x}_i = \delta_i = 0$ and consequently $x + \delta \geq 0$, i.e. $\delta$ is a feasible direction for $\overline{x}$. Hence, every element of $B(0, \varepsilon) \cap F_0 \cap \operatorname{Ker} A$ is a feasible direction for $\overline{x}$, and consequently any element of $F_0 \cap \operatorname{Ker} A$. Let $x \in \mathcal{X}$, and $v_1, \ldots, v_r$ a basis of $F_0 \cap \operatorname{Ker} A$ such that for every $i \in [r]$, $\|v_i\| = 1$. Let $\eta \in \mathbb{R}_+^*$ small enough such that $\forall i \in [r]$, $\overline{x} + \eta \delta_i \in \mathcal{X}$. We would like to show that for a well-chosen value of $\eta$, the following set of feasible directions for $x$, $\{\overline{x} + \eta v_1 - x, \ldots, \overline{x} + \eta v_r - x, \}$ is a basis of $F_0 \cap \operatorname{Ker} A$. Since $\overline{x} - x \in F_0 \cap \operatorname{Ker} A$, we consider $\beta_1, \ldots, \beta_r \in \mathbb{R}$ such that

$$\overline{x} - x = \sum_{i=1}^r \beta_i v_i.$$

Let $\alpha_1, \ldots, \alpha_r \in \mathbb{R}$. We have

$$\sum_{i=1}^{r} \alpha_i (\overline{x} + \eta v_i - x) = 0 \implies \underbrace{\left(\sum_{i=1}^{r} \alpha_i\right)}_{:=A} (\overline{x} - x) + \eta \sum_{i=1}^{r} \alpha_i v_i = 0$$

$$\implies \sum_{i=1}^{r} (A\beta_i + \eta\alpha_i) v_i = 0$$

$$\implies \forall i \in [r], \ A\beta_i + \eta\alpha_i = 0$$

$$\implies \underbrace{\forall i \in [r], \alpha_i = 0 \text{ or } \left(A \neq 0 \text{ and } \forall i \in [r], \ \frac{\alpha_i}{A} = -\frac{\beta_i}{\eta}\right)}_{(*)}.$$

By summing over $i$ in the last equality above, we get

$$(*) \implies \forall i \in [r], \alpha_i = 0 \text{ or } \left(A \neq 0 \text{ and } 1 = -\frac{1}{\eta} \underbrace{\sum_{i=1}^{r} \beta_i}_{B}\right)$$

The equality above is equivalent to $\eta = -B$. Hence, it suffices to take $\eta \neq -B$ and $\eta$ small enough to ensure that $\{\overline{x} + \eta v_1 - x, \ldots, \overline{x} + \eta v_r - x,\}$ is linearly independent and is a set of feasible directions for $x$, and consequently since all of these vectors are elements of $F_0 \cap \operatorname{Ker} A$, and there are $r = \dim F_0 \cap \operatorname{Ker} A$ of them, $\{\overline{x} + \eta v_1 - x, \ldots, \overline{x} + \eta v_r - x,\}$ is indeed a set of feasible directions for $x$ that is a basis of $F_0 \cap \operatorname{Ker} A$. $\qquad \square$

## D  Further Notes on Experiments of Section 6

**Data Generation.** The GPAs of candidates are generated using a uniform distribution in the interval $[2, 4]$, and the level of experience is also uniform in $\{1, 2, 3, 4, 5\}$. The results of Fig. 2 are from applying Algorithm 3 directly with the different sets $\mathcal{C}$ and $\mathcal{X}$. The MIP of Algorithm 2 is solved using Gurobi.

**Counter-Intuitive of Additional Constraints in $\mathcal{X}$.** One would naively expect that since $\mathcal{X}_{\text{experience}}$ is smaller than $\mathcal{X}_{\text{vanilla}}$, more data would be needed to make optimal decision in the vanilla setting, but that is not necessarily true. In Fig. 2, we see that in the high misspecification regime, more data is needed for the experience-constrained setting than the vanilla setting. In reality, the data needed depends on the geometry of the decision set $\mathcal{X}$ relative to the uncertainty set $\mathcal{C}$, as can be seen from Theorem 1 and Corollary 1.

**Note.** The hiring scenarios considered in this paper are stylized decision models intended to illustrate how data informativeness depends on task structure and prior uncertainty. While some formulations include group-based constraints (e.g., per-category quotas), these are not meant to prescribe or endorse any specific hiring policy.

