# OpenReview forum: "What Data Enables Optimal Decisions? An Exact Characterization for Linear Optimization"
_NeurIPS.cc/2025/Conference — NeurIPS 2025 poster_

### Official Review · Reviewer_6K14 · 2025-07-01

**Clarity:** 1
**Significance:** 1
**Originality:** 4
**Rating:** 2
**Confidence:** 2

**Summary:**

The goal is to choose  $x$ to minimize a loss function $L(x,\theta)$, where $\theta$ is a fixed but unknown parameter. Since $\theta$. is not known, they evaluate the loss at N points to estimate $\theta$. This paper considers the setting where the function $L$ is linear and the decision variable $x$ has to be selected from a polyhedral set. It is also assumed that the parameter $\theta$ comes from a given uncertainty set. The authors introduce the notion of a sufficient dataset, defined as a collection of $N$ objective functions that allow exact recovery of the true optimal solution. The paper proposes an algorithm for finding minimal sufficient dataset for this problem.

**Questions:**

- Can you provide another use case, which actually solves an LP optimization problem?

- What you want to show in Figure 2?

**Ethical Concerns:**

["NO or VERY MINOR ethics concerns only"]

**Final Justification:**

- I sincerely thank the authors for addressing my question during the rebuttal. Their answers helped me better understand the proposed technique. However, they also brought into focus what I see as the main weakness of the paper -- a lack of a clear problem definition with motivating example(s). Rather than starting with a concrete, well-motivated example, the paper opens with unrelated topics such as LLMs.   Because of this the paper fails to convince readers about the significance of the proposed approach.

- Similarly, the literature review does not connect clearly to linear programming. It is not addressed why  approaches -- such as predict-then-optimize or robust optimization -- are insufficient for the kind of problems they are tackling.

- Finally,  in their final remarks, the authors note: "We believe to be the first to address such a problem. Therefore, there are no baseline competitors to compare with." This statement also suggests that the problem as defined is not being addressed in practice, raising questions about its significance.

I believe the paper requires a revision providing a well-defined problem statement, clear motivation supported by a relevant example, and a focused discussion that avoids unrelated topics. However, given that this would require substantial restructuring, I must maintain my recommendation for rejection.

**Limitations:**

Yes

**Quality:**

2

**Strengths And Weaknesses:**

- The problem setting is difficult to understand and not clearly explained. Initially, the paper states that the parameter $\theta$ is not known. But later the concept of prior contextual knowledge is introduced.

- The example based on the Secretary Problem is difficult to follow. If $\phi$ represents resumes of the candidates, what does $\theta$ represent?   Also, in the Secretary Problem, the decision space increases as more candidates are interviewed, whereas the formal problem setup in Section 2 assumes a fixed-dimensional decision space $\chi \in \mathcal{R}^d$.

-  Algorithm 1 proposes a meta-algorithm that solves the LP optimization problem for every possible parameter value and stores all the unique solutions. This seems to be a trivial way to implement and I do not see the novelty. Indeed, if one could enumerate all possible solutions, the LP problem can be solved. Moreover, in most LP problems, the parameter space is continuous, making it unrealsistic to solve the LP for every possible parameter value $c$.

- In general, the paper lacks consistent focus. It begins with the motivation of defining the informativeness of a dataset for solving an optimization problem, but this idea is not mentioned in Sections 3 and 4. As a result, the narrative loses a clear goal  and seems disconnected from the motivation in the Introduction.

- I feel that the theoretical framework introduced in the earlier sections does not align well with the Secretary Problem use case. In the Secretary Problem, it is unclear what the basis vector of a solution would be. Additionally, LP solutions can be characterized by the extreme points (vertices) of the feasible region. So, if one stores all the vertices, the LP solution is guaranteed to come from this set. However, one cannot make a similar claim for the Secretary Problem. There is no guarantee that the best candidate will come from a predefined set of candidates.  Therefore, in my opinion, the Secretary Problem does not fit well with the problem formulation.

- There is only one experiment, not sufficient for empirical evaluation. Moreover, the setup is not clearly explained. it is not immediately clear what the authors want to highlight in Figure 2.

---

> ### Author Rebuttal · Authors · 2025-07-31
>
> There is a substantial misunderstanding of the concepts in the paper from the reviewer.  Several of the point mentioned by the reviewer are based on a wrong understanding of parts of the paper. In what follow we clarify the concepts and misunderstandings as a response to each point in the "Strengths and Weaknesses" and the Questions.
>
> On the points raised in "Strengths and Weaknesses":
>
> 1- The parameter $\theta$ is indeed not known, and this is consistent throughout the paper. Prior knowledge does not contradict that. $\theta$ is not fully known, but belongs to an uncertainty set, which models prior knowledge.
>
> The parameter $\theta$ is introduced for the first time in Line 32, directly as "an unknown parameter $\theta \in \Theta$". That is $\theta$ in unknown, but belongs to a set $\Theta$. Soon afterwards (Line 35) refers to the set $\Theta$ as prior knowledge: "The decision-maker can then use the observations along with their prior knowledge (set restriction $\theta \in \Theta$)".
>
> 2- In the example of the Secretary Problem, $\theta$ represents the values of the candidates. In the paragraph that introduces the example, in the first time where $\theta$ is mentioned (Line 53) it reads "Each candidate has an unknown value $\theta_i$".
>
> In the Secretary Problem, **it is not correct that the decision space increases as more candidates are interviewed**. In the Secretary Problem, the decision space $\cal X \subset \\{0,1\\}^d$ (Line 50), that is each decision is a binary vector, of which each coordinate correspond to one given candidate. The binary coordinate $x_i$ indicates whether candidate $i$ is hired or not: this is the decision in the problem. The decision space is therefore fixed here and does not change with more candidates interviewed. It is of dimension $d$, the number of available candidates (one dimension per candidate). An interviewing decision (a query $q_i$ revealing $\theta^\top q_i = \theta_i$, the value of candidate $i$) does not change the decision set $\cal X$ if this is what the reviewer means.
>
> 3- **It is not correct that Algorithm 1 consists of solving the LP for every possible value of c.** The paragraphs following Algorithm 1 describes how we implement the meta algorithm and Algorithm 2 provides a detailed version. The algorithm does not solve the original LP and certainly not for every value of $c$. It solves iteratively integer programs outputting the smallest basis spanning the relevant extreme directions of Theorem 2 and Theorem 1. It runs in at most $d$ iterations, where $d$ is the dimension of the problem as explicitly stated in Theorem 3. More precisely it runs in $k$ steps, $k$ being the dimension of the subspace spanned by relevant extreme directions.
>
> 4- Following the motivation and discussion around informativeness, we specify in the introduction, contribution paragraph, that "We study informativeness in the sense of being able to recover the task’s optimal solution" (Line 107).  Section 2 then mathematically models this question by the notion of a "*sufficient data set*" which is the focus of the paper. The subsequent sections focus on the technical resolution of this question, providing precise characterizations and an algorithm for identifying informative datasets in that sense.
>
> 5- We respond to the two parts of this remark separately:
>
> - Could the reviewer clarify what "the basis vector of a solution" means? If the reviewer's question is about basis vectors is the set of possible queries $\mathcal Q$ (Line 304), then these are $(e^1,\ldots,e^d)$ where $e^i_i=1$ and $e^i_j = 0$ for $i\neq j$ (indicator vectors). A decision $x=e_i$ corresponds to hiring candidate $i$  and no other candidate. A query $e_i$ corresponds to interviewing candidate $i$ as it reveals its value $\theta^\top e_i = \theta_i$. We explain these in paragraph introducing the Secretary Problem (Line 49), where we specify that the decision variable is binary shows which candidates are hired, and explains that interviewing a candidate means observing the coordinate of $\theta$ that specifies the score of the candidate (i.e. querying a vector from the canonical basis).
>
> - **The Secretary Problem is in fact mathematically a linear program** as explained in the introduction. It is solving $\min_{x \in \cal X} \theta^\top x$ where the constraint set is specified by linear constraints $\cal X \subset [0,1]^d$. This is also a well known result in the Secretary Problem literature on the Offline Secretary Problem. Its solutions are extreme point of this polyhedron (verifying $x \in \\{0,1\\}^d$) which corresponds to a hiring decision (which candidate to hire). **The best hiring decision will in fact be an extreme point.** In our paper, we introduce a relevant version of the multi-secretary problem that is offline, i.e. all candidates have already applied and the recruiter has to choose the best candidates.
>
>
> 6- Could the reviewer specify what is unclear about the experimental set-up and the discussion around Figure 2?
> The setup is that we have a set of candidate providing their GPA and a number of years of experience. We run our algorithm to decide which candidates to interview. The last paragraph of the experiment section specifically describes what we observe in Figure 2.  It illustrate how the optimal set of interviews (output of our algorithm) changes with the uncertainty and the hiring constraints.
>
> Response to Questions:
>
> 1- The reviewer's question seems to suggest that the Secretary Problem is not an LP. We would like first make it clear that the Secretary Problem is in fact an LP, as we discussed in response 5. We explicit that in the introduction and the experimental set up (Page 9, line before 340). It is also a very well known (and immediate) result in the Secretary Problem literature.
>
> To respond to the reviewer's question, another very relevant setting to our framework is the shortest path problem, which can be cast as a linear program. Assume for example that after an earthquake, emergency responders need to find the shortest path to an area to deliver important first aid resources, or launch a rescue operation, and can only check a limited number of routes. Assuming emergency responders have correlated data to the difficulty of each route, our algorithm can be used to specify which roads should be explored to find the shortest path for the delivery or/and rescue.
>
>
> 2- In Figure 2, we show how uncertainty and hiring constraint affect the optimal set to be interviewed. More uncertainty implies more candidates to interview and hiring constraints impact the interviews in a subtle way. The last paragraph of Section 5 discusses Figure 2 in details.

---

> > ### Comment · Reviewer_6K14 · 2025-08-01
> >
> > Thank you for your clarification. However, it is not enough to clarify me certain things. I am asking very specific questions to have better understanding:
> > - Can you provide the exact LP formulation of the Secretary problem?
> > - Can you clarify, what do you mean by Dataset in the context of standard LP formulation (1)?
> > - And what is the motivation of the proposed approach? I understand that the cost vector $c$ is not known. So, what does each query do? Does a query provide the perfect value of one coordinate of $c$?
> > - What is the distinction between $c$ and $\theta$? In Line 57, the objective function is: $\theta^\top x$, whereas in (1), it is $c^\top x$.
> >
> > - The description of Figure 2 writes "In the first row of Fig. 2, candidates fall into three groups: low scorers (never hired), high scorers (always hired), and mid scorers (interviewed)". I cannot see three groups in the Figure 2 and cannot a find a metric to evaluate the proposed approach.

---

> > > ### Author Response · Authors · 2025-08-01
> > >
> > > Thank you for the follow up. We are happy to further clarify in details the questions asked.
> > >
> > > 1) Following the notation and description in the introduction (Line 49-60). The Secretary problem can be cast as the following LP
> > >   $$\begin{align}
> > > 	\max_x & \\; \theta^\top x \\
> > > 	&0\leq x_i \leq 1, \\; \forall i\\
> > > 	&\sum_{i=1}^d x_i \leq k
> > > 	\end{align}$$
> > > 	where $\theta_i$ is the value of candidate $i$ and $k$ is the number of candidates we want to hire. Because of total unmodularity, the variable $x_i$ will always be $0$ or $1$ at optimum. Therefore $x_i\in\\{0,1\\}$ models whether we hire candidate $i$ ($x_i=1$) or not $(x_i=0)$. The constraints ensure we do not hire more than $k$ candidates. We can cast the problem as a minimization problem, by simply flipping the sign.
> > > 	$$\begin{align}
> > > 	\min_x & \\; -\theta^\top x \\
> > > 	&0\leq x_i \leq 1, \\; \forall i\\
> > > 	&\sum_{i=1}^d x_i \leq k
> > > 	\end{align}$$
> > > 	In the experiment we also explore the Secretary Problem with further hiring constraints, for example say the company wouldn't like to hire too many seniors, or too many juniors. This would correspond to adding the constraint $\sum_{i\in I_{\mathrm{junior}}} x_i \leq k_{\mathrm{junior}}$ and $\sum_{i\in I_{\mathrm{senior}}} x_i \leq k_{\mathrm{senior}}$ (see Section 5, $\cal X_{\mathrm{experience}}$) . This is also described in Line 51-52 of the introduction.
> > >
> > > 2) This is described in Lines 136–138 and earlier in Lines 30–36; we clarify further here.
> > > A dataset in the context of LPs is a set of evaluations of the objective function. A query $q \in \mathbb{R}^d$ yields the observation $c^\top q$. Since the cost vector $c$ is unknown, each query gives partial information about it. For instance, $q$ could be a feasible LP solution: we choose a decision and observe its cost. If $q = e_1 = (1,0,0,\ldots)$, then $c^\top q = c_1$, i.e., we observe the first coordinate of $c$. In the hiring problem, where $c \equiv \theta$ represents candidate values, this corresponds to observing candidate 1's value (i.e., interviewing them).
> > > Thus, a dataset ${\cal D} = \\{q_1,\ldots,q_N\\}$ yields observations $c^\top q_1,\ldots,c^\top q_N$, giving partial information on $c$.
> > > This matches our general definition of data informativeness in the introduction (Lines 30–39). When the loss $L$ is linear, $L(x,\theta) = \theta^\top x$, evaluating it at $(x,q)$ gives $q^\top x$, making the linear noiseless dataset definition consistent with the earlier general case.
> > > 3)
> > > - The motivation is to understand how data informs decisions, and more precisely what data enables optimal decisions. Classical work study the problem of what data is required to estimate an unknown parameter, rather to make a decision. Our work shows that the data needed for a decision in a specific task is different (and typically much smaller) than the one required to fully estimate the unknown parameter. In the well defined decision-making setting of linear programs, we are able to characterize exactly how the data requirement connects to the uncertainty ($\cal C$) and the task structure ($\cal X$). This shows how to chose data given a specific decision-making task and prior knowledge the decision-maker has.
> > > - As noted in point 2, each query $q \in \mathbb{R}^d$ yields $c^\top q$, a projection (linear combination of the $c_i$s)—i.e., an objective evaluation. The form of $q$ depends on the data collection setting. In hiring, $q \in \\{e^1,\ldots,e^d\\}$, where $e^i$ is the basis vector with $1$ in coordinate $i$ and $0$ elsewhere, so a query reveals the exact value of a candidate. In other contexts, $q$ may be other vectors—e.g., feasible decisions. In transportation, a decision-maker may choose paths and observe travel times, which amounts to measuring $c^\top q$ for path $q$ (e.g., the shortest path problem is a linear program).
> > > 4) $c$ is the same as $\theta$. In the introduction, we use the standard parameter notation $\theta$ when discussing general informativeness in general problems. When we specify the set-up to linear programs, and the unknown parameter to the cost vector, we use the notation $c$ as it is standard for cost vectors. In Line 135, we explicitly say that $c$ is $\theta$ in the linear setting "...uncertainty set $\cal C \subset \mathbb{R}^d$, which captures prior information on $c$ (these are $\theta$ and $\Theta$)."
> > > 5) In Figure 2, candidates with higher GPA and experience tend to score better (up to noise $\varepsilon$). The lines show combinations of GPA and experience that yield the same score (ignoring noise). Moving northeast—perpendicular to the lines—indicates stronger candidates. Those in the southwest are not interviewed due to clearly low scores; those in the northeast are also skipped since their high scores are clear. Only the red ones in the middle "gray zone", for which features indicated neither too low or too high scores are uncertain due to noise, are interviewed.
> > >
> > > We hope this clarification helps and will be considered in reassessing the paper.

---

> > > > ### Comment · Reviewer_6K14 · 2025-08-02
> > > >
> > > > Thank you for providing the LP formulation. I now realize it closely resembles the well-known Knapsack problem.
> > > >
> > > > The problem definition is estimating the cost parameter $c$, knowing that it originates from an uncertainty set. The paper aims to establish that  making optimal decisions requires less data than making accurate predictions. But, the problem of learning cost parameters for linear optimization problems under uncertainty has already been the subject of significant research, including by Berthet et al. (2020), Elmachtoub and Grigas (2022), and Tang and Khalil (2024). A comprehensive survey of this line of work is available in Mandi et al. (2024). The current paper makes no attempt to position itself within this body of literature, a major weakness.
> > > >
> > > > I would also recommend evaluating the proposed approach using experimental setups from the above works. Additionally, the current evaluation lacks a proper metric, readers are expected to rely on visual intuition from the figures, which is not ideal for a NeurIPS-level submission.
> > > >
> > > > Further, the paper would benefit from a more compelling motivating example. The use of the secretary problem is weak, because of  the recommendation that high scorers would be hired without interviews. This assumption is unrealistic -- organizations do not hire a person without an interview, regardless of their credentials.
> > > >
> > > > *References*
> > > >
> > > > - Berthet, Q., Blondel, M., Teboul, O., Cuturi, M., Vert, J. P., & Bach, F. (2020). Learning with differentiable pertubed optimizers. Advances in neural information processing systems, 33, 9508-9519.
> > > >
> > > > - Elmachtoub, A. N., & Grigas, P. (2022). Smart “predict, then optimize”. Management Science, 68(1), 9-26.
> > > >
> > > > - Tang, B., & Khalil, E. B. (2024, May). Cave: A cone-aligned approach for fast predict-then-optimize with binary linear programs. In International Conference on the Integration of Constraint Programming, Artificial Intelligence, and Operations Research.
> > > >
> > > > - Mandi, J., Kotary, J., Berden, S., Mulamba, M., Bucarey, V., Guns, T., & Fioretto, F. (2024). Decision-focused learning: Foundations, state of the art, benchmark and future opportunities. Journal of Artificial Intelligence Research, 80, 1623-1701.

---

> > > > > ### Author Response · Authors · 2025-08-03
> > > > >
> > > > > The reviewer brings about three new points in the follow-up: connection to the predict-then-optimize literature, the choice of experiments in the paper, and the relevance of the chosen example. We detail below each point.
> > > > >
> > > > > - **The “predict-then-optimize” literature addresses a fundamentally different problem than our paper.** The authors are well familiar indeed with this line of work. The works cited (e.g., Berthet et al., Elmachtoub and Grigas, Tang and Khalil) focus on learning or *estimating* a cost vector from data in order to make good decisions. That is *data is given and the goal is to make the best use of it*. In contrast, our work *does not attempt to estimate the cost vector*. Instead, it asks the question: *What information/data must be acquired to find an optimal decision?* This is *not an estimation* problem, but a *data sufficiency* problem for decision-making—a fundamentally different question, both in motivation and technical treatment.
> > > > >
> > > > >   We would also like to emphasize that **it is not correct that "The problem definition is estimating the cost parameter, knowing that it originates from an uncertainty set."** Our paper is about understanding *the data part of the question*: what data enables optimal decisions.
> > > > >
> > > > > - **On the Experimental Setup and Metrics.** Our paper is a theoretical paper. The main contribution is the proven characterization: an exact quantification of what data enables optimal decisions, and the algorithm to compute it. This is the first result to quantify data requirements for decision-making, in a non-adaptive way, to the best of our knowledge. The goal of the experiment is to illustrate visually our theory in an application, *not to benchmark against other algorithms*. In fact, there are no algorithms to benchmark here.
> > > > >
> > > > >   As the reviewer is aware, our paper is submitted to NeurIPS in the Primary Area "Theory". NeurIPS is rich in both empirical papers and theoretical papers. The latter often do not need experiments, which are beyond the point of the paper. The reviewer understands, as is standard in theoretical NeurIPS papers, that the lack of such extensive numerical experiments is not inherently a weakness when the theoretical results do not warrant or naturally involve any numerics or benchmarks.
> > > > >
> > > > >   As to the absence of "metrics" in our experiments, the goal of the experiments is to illustrate the structural insights obtained from our theoretical results. It is not to benchmark performance. In our theorems, sufficiency of a dataset is characterized as a geometric condition relating the structure of the decision task and the uncertainty set to the data requirement. The experiment illustrates this geometric relationship in a practical application. It allows us to observe visually how changes in the task structure and uncertainty impact data requirements. There is no ambiguity in interpretation: the visualizations directly show the outcome of applying our exact characterization. Introducing extraneous metrics would not serve this purpose.
> > > > >
> > > > >
> > > > > - **On the Hiring Example.**
> > > > > 	The hiring example is designed to illustrate how the theoretical framework captures the interplay between uncertainty and task constraints in determining data needs. The qualitative features it exhibits—such as partitioning candidates into always-hired, never-hired, and those requiring interviews—should be understood in general in the context of a ranking and selection problem. Naturally, without further "practical" constraints, queries informing a ranking decision should naturally focus on the candidates in the "boundary". This is what our algorithm correctly identifies. Furthermore, with more uncertainty, the "boundary" (and candidates to interview) grows larger: this is again what the algorithm identifies in Fig 2. Further insights are observed when adding constraints to the task, illustrated in Fig 2.
> > > > >
> > > > > 	Comments about the realism of the scenario do not pertain to the purpose of this illustrative example, which is to convey intuition for the theoretical results, not to model practical hiring practices.
> > > > >
> > > > > 	As an aside, even with this simplistic example, the obtained result is not trivial. Several universities hire PhD students within this structure: they identify a set of "top" candidates that they admit outright, and then interview borderline candidates to refine their choice. That said, the goal of our example is not to condone or recommend any hiring practice; it is simply to illustrate how data informs decisions and which data points matter most in a simple example.

---

> > > > > > ### Comment · Reviewer_6K14 · 2025-08-04
> > > > > > **Response**
> > > > > >
> > > > > > Thank you for your response; it made a few things clearer for me. I  have reassessed the paper with your explanations in mind.

---

> > > > > > > ### Author Response · Authors · 2025-08-04
> > > > > > >
> > > > > > > We're glad our response helped clarify the paper, and we appreciate the reviewer’s reassessment. Thank you for the follow-up and for engaging thoughtfully in the discussion. We would be happy to address any further questions that may arise.

---

> > > > > > > ### Author Response · Authors · 2025-08-08
> > > > > > >
> > > > > > > Dear reviewer,
> > > > > > >
> > > > > > > Thank you again for engaging in the discussion and the follow-up questions.
> > > > > > > We appreciate you reassessing the paper after our responses and clarifications.
> > > > > > >
> > > > > > > As the discussion period is coming to an end, we wanted to indicate that we did not see (in our console) an update to the scores with your reassessment, as mentioned in your last response.
> > > > > > > We just wanted to indicate that in case it was missed or was not properly accounted for by the system.
> > > > > > >
> > > > > > > We thank you again for the reviewing effort.

---

> > > > > > > > ### Comment · Reviewer_6K14 · 2025-08-09
> > > > > > > > **Clarification: I kept my score unchanged**
> > > > > > > >
> > > > > > > > I apologize for the confusion. I have re-evaluated the paper based on the response to my questions and to other reviewers' questions. My score remains unchanged, and I still recommend rejection based on the following key points:
> > > > > > > >
> > > > > > > > - While I appreciate Theory papers in Neurips, a theoretical paper for an AI/ML conference like Neurips should provide clear motivation for the proposed technique with some discussion on practical use cases. That would make the readers appreciate 'why' the proposed theory is significant.  The paper has proposed a novel technique in the "how" part,  but the "what" and "why" remain ambiguous. The lack of a clear motivational use case makes the paper's overall motivation unclear.
> > > > > > > >
> > > > > > > > - I find the literature review to be a significant weakness as a Theory paper. The paper discusses Active Learning, Informativeness Theory, and Robust Statistics, but they are not related to Linear Optimization, the main focus of the paper. A more relevant body of literature is "predict-then-optimize", which also deals with linear optimization under uncertainty. This body of literature is not discussed. While the problem formulation here may be slightly different, as explained in the rebuttal, a good Theory paper would acknowledge and distinguish itself from this related work.
> > > > > > > >
> > > > > > > > - The paper's readability and presentation in the Algorithm Section require further improvement. The core theoretical contribution is hard to follow. For example, the paper's main body presents Algorithm 1, which appears to solve an LP optimization for every possible parameter vector $c$. Whether the rebuttal clarifies that Algorithm 2 is the one actually being solved, the authors should have provided Algorithm 2 in the main text and provided a clear explanation of the optimization problem solved in Algorithm 2. Without this, the readers cannot fully appreciate the proposed technique.
> > > > > > > >
> > > > > > > > - Finally, I agree that a Theory paper can be accepted without an Experimental Evaluation. But if the paper contains Experimental Evaluation,  it must be sound. Experimental evaluation without a baseline competitor is a major weakness. Moreover, being a Theory paper, it should provide a suitable evaluation metric for the problem being studied. There should be a metric to evaluate the effectiveness of the proposed technique.
> > > > > > > >
> > > > > > > > While some of these issues can be addressed as the authors demonstrated in the Rebuttal, addressing them would require substantial revisions to the paper. In journal terminology, the revisions I highlighted would constitute a major revision. Given that conference papers are accepted with minor revisions, my decision is to reject this paper in this edition.

---

### Official Review · Reviewer_At1w · 2025-07-03

**Clarity:** 3
**Significance:** 3
**Originality:** 2
**Rating:** 4
**Confidence:** 3

**Summary:**

This paper investigates when a dataset contains sufficient information to recover the optimal solution to a linear program under cost uncertainty. The authors provide a sharp geometric characterization of task-relevant directions and propose an algorithm to construct minimal sufficient datasets. The work offers a principled perspective on task-aware data selection.

**Questions:**

See Weaknesses.

**Ethical Concerns:**

["NO or VERY MINOR ethics concerns only"]

**Final Justification:**

I think the authors address some of my key concerns

**Limitations:**

yes

**Quality:**

2

**Strengths And Weaknesses:**

**Strengths:**
- The paper addresses a fundamental and important question at the core of data-driven decision-making.
- The theoretical contributions are a key strength. The authors establish a clean formalism and derive their results with rigorous proofs. The connection established in Thm. 2 is particularly deep and non-trivial, creating an elegant bridge from pure theory to practical application.
- The paper is well-written, and its arguments are clear.

**Weaknesses:**

- The proposed algorithm relies on iteratively solving an MILP. While this is a standard and powerful approach, MILPs can be computationally intensive, and the paper lacks a detailed discussion of the algorithm's practical scalability.
- The focus on a noiseless setting is a notable limitation. While Prop. 3 offers a valuable local robustness guarantee; the concept of "sufficiently small" noise would be more tangible with concrete and practical examples to illustrate its scope.

---

> ### Author Rebuttal · Authors · 2025-07-31
>
> We thank the reviewer for their insightful remarks.
>
> - As specified by the reviewer, MILPs can be computationally intensive, though they tend to perform well in practice when the problem is well structured. We mention in line 333 that the number of variables and constraints in the MILP solved at each iteration scales linearly with the dimension of the problem and the number of constraints. Hence, the size of the MILPs scales reasonably well with the size of the problem. To further substantiate our claim, we ran our algorithm for large instances of two relevant problems. The first is the vanilla hiring problem, where there were 3000 candidates, and 32 interviews were prescribed by the algorithm, which completed after roughly 8 minutes of computation. Similarly, we tested the algorithm on a shortest path problem with 1291 edges and 604 nodes. The algorithm prescribed 7 directions to query and finished running after 15 minutes. These experiments were conducted on a regular laptop, which illustrates that the practical computational complexity scales reasonably with the problem size. However, it is true that for significantly larger instances (in the order of $>10^5$ variables), solving the MILPs may become more challenging.
> - This is indeed a valid point. In Proposition 3, we prove that when the noise level is low enough, then a sufficient dataset still recovers the optimal solutions. In order to quantify the change in optimality beyond what is covered by proposition 3, we extend this result: when the noise amplitude exceeds the threshold $\kappa$, the normalized optimality gap scales linearly with the noise magnitude. More precisely, we can prove that if $c_{True}$ is the true cost vector, $o$ is the vector containing query observations, and $\varepsilon$ is a real-valued vector, $\hat{x}(o+\varepsilon)$ the decision inferred by our algorithm given noisy observations $o+\varepsilon$, and $x^\star$ the optimal decision for the cost $c_{True}$, then we have $c_{True}^\top \hat{x}(o+\varepsilon) - c_{True}^\top x^\star = O(||\varepsilon||)$ if $||\varepsilon||$ exceeds some threshold $\kappa > 0$, and $c_{True}^\top \hat{x}(o+\varepsilon) - c_{True}^\top x^\star = 0$ otherwise. This demonstrates that, in the worst case, the error introduced by our algorithm scales linearly with the noise amplitude.

---

> > ### Comment · Reviewer_At1w · 2025-08-06
> >
> > Thank you for your response; it made a few things clearer for me.

---

> > > ### Author Response · Authors · 2025-08-08
> > >
> > > Thank you for the efforts spent on reviewing our paper and for the insightful questions. We are glad our response was useful!

---

### Official Review · Reviewer_jqbm · 2025-07-09

**Clarity:** 3
**Significance:** 3
**Originality:** 3
**Rating:** 5
**Confidence:** 3

**Summary:**

This paper addresses the fundamental question of how informative a dataset is for solving decision-making tasks, with a focus on linear optimization under uncertainty. It introduces a geometric framework to determine when a dataset is sufficient to recover the optimal decision, given partial information about the cost vector. The authors provide exact characterizations using relevant extreme directions and develop an efficient algorithm to construct minimal sufficient datasets. Their results demonstrate that small, carefully chosen datasets can fully determine optimal solutions, providing a principled foundation for task-aware data selection and efficient data acquisition in settings such as hiring or resource allocation.

**Questions:**

- I am wondering whether the framework can be extended to making-decision problems in RKHS space, Bayesian optimization or nonlinear programming? What are the challenges?

**Ethical Concerns:**

["NO or VERY MINOR ethics concerns only"]

**Limitations:**

Yes

**Quality:**

3

**Strengths And Weaknesses:**

Strengths:

- The paper considers an important question of ML

- The paper provides conditions for when a dataset is sufficient to recover optimal decisions in linear programs. This sharp geometric formulation, based on relevant extreme directions.

- The fact that the span of relevant directions for dataset sufficiency can equivalently be expressed as the span of differences between optimal solutions under different cost vectors in the uncertainty set is interesting. From this, the authors develop a practical and efficient algorithm to construct minimal sufficient datasets.

Weaknesses:

-  The theory assumes that the decision-maker can query arbitrary directions and observe the exact inner product. This may be a strong assumption for making decision problems I have known.

-  The paper’s results apply to linear optimization where the constraint set $X$ is fully known and fixed. This can limit the generality and applicability of the framework to nonlinear programming.

---

> ### Author Rebuttal · Authors · 2025-07-31
>
> We thank the reviewer for the time spent carefully reviewing our paper and for their insightful remarks. Below are our responses to the raised points in Weaknesses:
>
> - Our theory does not assume that the decision maker can query arbitrary directions. In section 4, we specify that in practice, the possible set of queries can be constrained to a set $\mathcal Q$, and show how our algorithm can be adapted to take this constraint into account. To the reviewer's remark, we can refer to this section earlier in the paper to make that clearer.
>   As for observing the exact inner product, in fact this is a limitation in the case where measurements are noisy. Proposition 3 addresses this limitation by showing that a sufficient dataset remains sufficient with noisy measurements as long as the noise is small enough. That is, in a setting with noisy observations (noisy inner product observations), our algorithm would still output a sufficient dataset as long as the noise is small enough.
>
> - Our theory can be adapted to model the case when $\mathcal X$ is not fixed. Indeed, for example suppose $\mathcal X =\\{x\in \mathbb R^d, \\; Ax=b,\\; x\geq0 \\}$. Uncertainty in $\cal X$ can be modeled by either uncertainty in $b$ or in $A$. Our theory already captures uncertainty in $b$. In fact, by considering the dual of the linear program, the new cost vector becomes $b$, and our algorithm can be applied with queries on $b$ rather than on $c$.
>   However, the case of uncertainty in $A$ or jointly in $b$ and $c$ is not directly addressed by our theory, and that remains a limitation. This would be an interesting direction of work.
>
> As for your question: The main bottleneck to adapt our theory to more general settings (RKHS, Bayesian optimization, nonlinear optimization...) is the structure of the decision set, and most importantly, the structure of the boundaries between optimality regions. The fact that optimality regions in the case of linear programming have a polyhedral structure (polyhedral cones) allows us to have a clean characterization of which queries allow us to distinguish them efficiently. This theory can be adapted by generalizing the notion to more complicated boundaries between optimality regions of other optimization problems. One promising venue for that is leveraging the theory behind Parametric Programming, which studies such optimality regions.

---

### Official Review · Reviewer_zZVt · 2025-07-17

**Clarity:** 3
**Significance:** 3
**Originality:** 3
**Rating:** 5
**Confidence:** 3

**Summary:**

The paper proposes a framework to identify the minimal sufficient dataset required to solve a linear program (LP) where the cost vector is uncertain. The authors provide a geometric characterization for data sufficiency based on the problem's constraints and the uncertainty set. They further develop an algorithm based on this theory to construct such a dataset for offline data collection settings.

**Questions:**

1. How does the proposed methodology relate to sensitivity analysis for parametric and robust linear programming? What key theoretical contributions are novel beyond applying post-optimal analysis tools to a data sufficiency problem?

2. How should a practitioner realistically define the uncertainty set $\mathcal{C}$ for problems where the magnitude of unobserved factors is unknown beforehand?

3. How to revise the experiments to better demonstrate the usefulness of the proposed methodology by addressing weaknesses points 3 and 4?

**Ethical Concerns:**

["NO or VERY MINOR ethics concerns only"]

**Final Justification:**

While I acknowledge the paper is technically sound within its defined paradigm, I remain unconvinced of its practical value proposition. For these reasons, my rating will stay the same as Broadline Accept; the work is theoretically interesting, and I'm not against its acceptance, but its practical impact remains an open question.

Updated: After reviewing all the final responses the author provided, I decided to change my score to Accept. I believe it is a solid paper worth accepting.

**Limitations:**

NA.

**Paper Formatting Concerns:**

Page 4, line 173, 'formulate' should be 'formulates'.

Page 5, line 195, 'an infinite sets'.

Page 9, line 346, 'a similar pattern arise'.

**Quality:**

3

**Strengths And Weaknesses:**

Strengths:

1. The paper is well-written and easy to read.  The ''data sufficiency'' question in the offline setting is novel to me. The theoretical results are written clearly with a good pace.

2. The theoretical support is rigorous. The paper provides necessary and sufficient conditions for data sufficiency, grounding the problem in the polyhedral geometry of the LP's decision space and uncertainty set.


Weaknesses:

1. $\textbf{Overlap with sensitivity analysis.}$ The paper's main theoretical contribution appears to have overlap with established concepts in the literature on sensitivity analysis for robust LP, which is not sufficiently discussed or cited. The core theoretical mechanism of identifying "relevant directions" ($\Delta(\mathcal{X}, \mathcal{C})$) by checking whether the uncertainty set $\mathcal{C}$ intersects with the faces of the LP's optimality cones is a fundamental tool to post-optimal analysis. The paper's framework seems to be a re-application of these well-known principles to answer a "data sufficiency" question, rather than a fundamentally new theoretical discovery.

2. $\textbf{Uncertainty sets $\mathcal{C}$.}$  The main weakness of the proposed methodology is the framework's dependence on a pre-defined uncertainty set $\mathcal{C}$. The method's usefulness depends on the ability to specify this set, or equivalently, knowing the noise level before data collection. This is challenging in practice, as the magnitude of unobserved, "Game-Changing" factors is often what data collection aims to discover (see point 3 for an example). If $\mathcal{C}$ is defined too conservatively to be safe, the method's recommendations can become trivial (i.e., collect all data).

3. $\textbf{Oversimplfied data example.}$ The paper's data example (hiring) is based on unrealistic assumptions. It implies that unobserved factors (e.g., personality, collaboration skills) have a bounded, non-decisive impact on extreme candidates (which corresponds to a high noise level regime in the paper). As stated in 2, if the unobserved factors are decisive, the method will be trivial to recommend collecting all the data.

4. $\textbf{Experimental setup inconsistent with proposed methodology.}$  There is a mismatch between the paper's theoretical treatment of noise and its experimental setup. Proposition 3 shows that with a fixed uncertainty set $\mathcal{C}$, using data sufficiency is robust to small measurement noise $\varepsilon$. However, the experiment takes a different approach: it constructs the uncertainty set $\mathcal{C}$ itself by using a known, underlying "noise level" parameter $\eta$. This experimental design avoids the main practical issue of how a user would define $\mathcal{C}$ without prior knowledge of $\eta$, thus failing to demonstrate how the framework would be applied in reality.

---

> ### Author Rebuttal · Authors · 2025-07-31
>
> We thank the reviewer for the efforts spent carefully reviewing our paper and providing valuable comments. Below, we address and answer in details the reviewers comments.
>
> **(Pt 1 and Q1) Connections to Sensitivity Analysis, Parametric and Robust Linear Programming.** The reviewer’s recommendation to connect our work to this literature is a very good point. Our literature review indeed focused more on the "data part" of the question rather than the specific aspects related to linear programming. Below we clarify how our work connects with this literature. We can include this discussion in the revised version of the paper, should the reviewer find it useful.
>
> The goal in this stream of literature is to understand how small (sensitivity analysis, **SA**) or large (parametric programming, **PP**) changes in the LP parameters ($A$, $b$, $c$) affect the optimal value and solution (Ward and Wendell 1990). Most relevant to our setting are changes in $c$ or $b$ (since all our theory also applies to uncertainty in $b$ via the dual LP). More specifically, a typical goal is (roughly) to characterize the function $\lambda \to \arg \min_{x \in \cal X} c(\lambda)^\top x$ when the cost vector changes according to some parametrization $\lambda \mapsto c(\lambda)$ within a predefined uncertainty set $\mathcal{C} = \{c(\lambda): \lambda \in \Gamma\}$ (Gal and Nedoma 1972; Saaty and Gass 1954). Typically within a lower dimensional, linear parametrization.
> Our goal is different: we aim to understand what data sets (queries of the cost vector) allow recovery of the optimal solution of the LP. That is, we study how _partial information_ about the cost vector _informs_ the optimal solution, rather than how _changes_ in the cost vector _alter_ the optimal solution.
>
> The fundamental connection is that both goals involve understanding the relationship between the cost vector $c$ and the optimal solution. Naturally, studying this interplay involves similar mathematical objects: optimality cones (analogous to critical regions in SA and PP, although slightly different) and feasible directions that change the optimality of a solution and the ones preserving it.
>
> Although our technical results involve similar objects, none of our results or proofs use existing results from the SA or PP literature, nor are they direct consequences of them. Below we further detail the differences in technical goals and proofs.
>
> At the technical level, a typical question in SA is to identify the _distance to the boundary of the critical region_ (or optimality cone) along a certain direction—i.e., how much change in that direction preserves optimality. In PP, a typical goal is to _characterize all critical regions_ (or optimality cones) along some dimension, as this allows one to construct the function $\lambda \to \arg \min_{x \in \cal X} c(\lambda)^\top x$ which is piecewise constant over the (potentially exponentially many) regions.
> Our technical goal is different. Data sufficiency involves computing the _**subspace spanned by the relevant directions**_ rather than explicitly computing the optimality cones, the distances to their boundaries or the directions themselves. Relevant directions here are not all directions that change the solution, but only the most "informative" ones—those orthogonal to the cone faces—as these suffice to distinguish cones for our objective. There may be exponentially many such directions, data sufficiency does not require computing all of them, but only the *subspace they span*. To the best of our knowledge, this subspace has not been studied before. Our Theorem 2 is a fundamental result that allows computing it by solving at most $k$ MIPs, where $k \leq d$ is the dimension of the subspace, rather than exhaustively enumerating all (exponentially many) relevant directions/optimality cones.
>
> Although we did not leverage techniques and results from SA and PP in our current paper, some of the methods developed in this rich literature for handling these mathematical objects could be useful in our context and future extensions. For instance, PP has studied problems beyond the LP case (MIPs, non-linear problems), and we could be inspired by how notions of optimality cones and directions are extended in those non-linear settings. We thank the reviewer for highlighting this connection.
>
> Citations:
> (Ward and Wendell 1990) Approaches to sensitivity analysis in linear programming
>
> (Gal and Nedoma 1972) Multiparametric Linear Programming
>
> (Saaty and Gass 1954) Parametric Objective Function
>
> **(Pt 2 and Q2) Uncertainty set $\mathcal{C}$.** This is an important point: how should $\mathcal{C}$ be chosen in practice? There are essentially two components in the design of the uncertainty set:
>
> *Expert knowledge.* For instance, in the hiring problem, candidate scores are non‑negative and may satisfy ratio bounds (e.g., no candidate is more than ten times better than another). In a transportation problem, edge costs are positive and upper bounded. Similarly, one may know the sign of the correlation between the score and certain features (e.g., $\alpha_k \ge 0$ or $\alpha_k \le 0$ for some feature $k$), while the “noise level” ($\eta$ in the definition of $\mathcal{C}$ on page 9) remains unknown.
>
> *Past data.* In addition to expert knowledge, historical data lets us build practical, data‑driven uncertainty sets via high probability confidence sets with classical statistical tools.
>
> Let us illustrate it in the hiring example. Past data would correspond to previously interviewed candidates (from earlier rounds, separate from the current pool). This means the decision maker has a set of previously observed feature vectors and associated values, $(c'_i, \phi'_i)$ for $i\in[N]$.
>
> Let $X\in\mathbb{R}^{N\times d}$ collect past features in its rows, with $X_i^\top=\phi_i'$. Under the linear model $c = \alpha^\top \phi + \epsilon$ with i.i.d. Gaussian errors $\epsilon\sim\mathcal{N}(0,\sigma^2)$, the least‑squares estimator and empirical noise variance are
>   $$
>   \hat{\alpha}=(X^\top X)^{-1}X^\top c', \qquad
>   \hat{\sigma}^2=\frac{1}{N-d}\sum_{i=1}^N (c_i'-\hat{\alpha}^\top \phi_i' )^2.
>   $$
>   A $(1-\gamma)$ confidence interval for each coordinate $\alpha_j$ is
>
>  $$
>   \hat{\alpha}_j \pm t _{1-\gamma/2,N-d} \sqrt{\hat{\sigma}^2(X^\top X)^{-1} _{jj}}
>   $$
>
>   where $t_{1-\gamma/2,N-d}$ is the Student‑$t$ distribution quantile. Moreover,
>   $$
>   \frac{(N-d)\hat{\sigma}^2}{\sigma^2} \sim \chi^2_{N-d},
>   $$
>   which yields a high‑probability upper bound on $\sigma$ (and hence on the noise parameter $\eta$) via the corresponding $\chi^2$ quantile. These provide then data-driven choices for the parameters ($\eta$, $l,u$) of the uncertainty set $\cal C$ of page 9.
>
> This illustrates *one* approach to designing an uncertainty set. The *robust‑optimization literature offers many ways to build $\mathcal{C}$ from classical statistical tools*, each with different properties. More recently, conformal prediction provides distribution‑free confidence sets built from any predictive model (e.g., a neural network) to design an uncertainty set.
>
> **(Q3) As a response to Q3**, we can redesign the experiment in light of these observations by generating past data (a couple of candidates previously interviewed) and design $\eta,u,l$ directly from that data. Then see how the data requirement changes as a function of the past data size rather than hand-chosen noise levels $\eta$. This would give a practical case of uncertainty sets.
>
> **(Pt 3 and Q3) More on the hiring data example: impact of non-decisive features and experimental set-up.**
> We would like to emphasize that our algorithm does not need any assumption on the impact of unobserved factors (features). It simply determines precisely the data requirements based on whatever prior knowledge (uncertainty set) provided to it.
> If the unobserved features are highly decisive, the algorithm will naturally recommend extensive data collection—which is exactly the correct recommendation in that case. In this case, it is fundamentally impossible to reduce the extensive data requirement, given the unobserved factors. The algorithm provides the necessary data requirement given the decision maker’s level of knowledge, whether that knowledge is rich or limited (i.e., whether uncertainty is low or high).
>
> In our specific example, we used only two features to visually illustrate the results in a 2D plot. A realistic application would incorporate all available features. We do not impose any assumption about their informativeness; this is instead captured by $\eta$, the noise amplitude. For different values of $\eta$, we evaluate the resulting data requirements which naturally increases with the level of uncertainty.
>
> **(Pt 4 and Q3)  Experimental setup and misunderstanding of two different noise notions.**
> In point 4 of the reviewer’s comments, there appears to be confusion between two distinct notions of noise used in the paper. The noise in Proposition 3 refers, as the reviewer correctly notes, to *measurement noise*. In the experiment, this would correspond to observing an *approximate score* $c_i$ when interviewing candidate $i$, rather than the exact score. Proposition 3 states that a sufficient dataset under exact measurements remains sufficient in the presence of small measurement errors.
>
> In the experiment section, however, what we refer to as noise ($\epsilon$ and $\eta$) is actually the _misspecification level_ of the linear model: measuring how informative the features $\phi$ are (how well they predict the scores). This captures the level of uncertainty in this particular design of the uncertainty set and is a completely different concept from the measurement noise of Proposition 3. We should have explicitly referred to it as the misspecification level or uncertainty level to avoid confusion.
>
> Regarding the knowledge of $\eta$ in the design of the uncertainty set, we detail how it can be estimated using past data in Pt2.

---

> ### Comment · Reviewer_zZVt · 2025-08-03
> **Response to Rebuttal by Authors**
>
> Thank you for your detailed response. Your clarifications regarding the distinction from sensitivity are well-taken and helpful. I recommend integrating these discussions into the revised manuscript.
>
> However, your response has confirmed my primary concern regarding the paper's real-world significance.
>
> The proposed solution for defining the uncertainty set C, by using statistical confidence sets from past data, exposes a fundamental, philosophical weakness. By offloading the most challenging part of the problem (characterizing uncertainty) to a standard statistical procedure, the robust optimization framework becomes a deterministic step applied to a probabilistic basis. This raises the question: why is this two-stage, hybrid approach superior to a more integrated, fully statistical framework (e.g., Bayesian decision theory) that handles both uncertainty estimation and decision-making coherently under a single probabilistic roof?
>
> The simplified examples emphasize this concern. For the methodology to work smoothly, the problem must be reduced to a "toy" version where the uncertainty is manageable and clearly bounded. This does not convince me of the method's usefulness for complex problems, where the main challenge is understanding the nature of the uncertainty itself.
>
> Therefore, while I acknowledge the paper is technically sound within its defined paradigm, I remain unconvinced of its practical value proposition. For these reasons, my rating will stay the same as Broadline Accept; the work is theoretically interesting, and I'm not against its acceptance, but its practical impact remains an open question.

---

> > ### Author Response · Authors · 2025-08-04
> >
> > Thank you for the follow-up and once again for the insightful and well-thought-out comments.
> >
> > We are pleased that the discussion on sensitivity analysis was helpful, and we will incorporate it into the revision. It will indeed be a valuable addition to the paper.
> >
> > The main concern raised by the reviewer fundamentally centers on **set-based modeling of uncertainty** versus **distribution-based modeling of uncertainty**—a question that has been extensively discussed in the literature, with a rich body of work outlining the benefits and limitations of each. In particular, the robust optimization literature (now numbering thousands of papers) has long emphasized the advantages of set-based uncertainty, while the stochastic optimization literature has traditionally argued for distribution-based modeling, including Bayesian approaches.
> >
> > Our paper does not take a position in favor of one approach over the other; rather, it adopts the well-established set-based modeling framework—an approach that is not novel to our work and comes with both advantages and limitations.
> >
> > We are preparing a more detailed response outlining the benefits and limitations of each model, from the literature around both. We also acknowledge that the example provided in our previous response may have been too narrow (a single example) and not fully representative of the diverse techniques available for designing uncertainty sets, nor of the arguments for preferring them over distribution-based models in certain contexts.
> >
> > We will post this more detailed response shortly, and we thank the reviewer again for their careful reading and thoughtful feedback.

---

> > ### Author Response · Authors · 2025-08-07
> > **On the practicality of set-based uncertainty modeling**
> >
> > **Part 1/2**
> >
> >
> > Following on our last response, we explain below how past literature established uncertainty sets as a practical approach to modeling uncertainty. There is a vast literature on robust optimization that argues to the merit of such an approach. An important number of these papers have successfully applied this approach in real-life problems as we detail next. In short, distributional-based approaches are intuitively more granular in capturing uncertainty. However, they are limited in their practicality in several settings, due to tractability challenges and relying more heavily on assumptions for their validity. Set-based uncertainty modeling, on the other hand, captures uncertainty in a more "summarized" way, at the risk of being more conservative. This comes with the important benefit of tractability and less reliance on assumptions.
> >
> > Ultimately, both modelings have their advantages and limitations. Our work does not aim to argue that either set-based or distribution-based uncertainty is inherently more useful. Rather, we adopt set-based uncertainty as a modeling choice, motivated by its practical advantages in our setting. We now elaborate on some elements of the distinction between set-based and distribution-based uncertainty, which is well-studied in the literature.
> >
> > - **Assumptions.** Perhaps one of the most straightforward differences between set-based and distribution-based uncertainty lies in their underlying assumptions. Set-based approaches typically rely on milder assumptions, as they do not require a fully specified probabilistic model of uncertainty. Instead, uncertainty is captured through bounds or confidence sets that are valid for a general class of distributions (such as with a given finite moments, or a given support). This makes set-based methods particularly attractive when we have limited information on the underlying model. In contrast, distribution-based approaches assume that the uncertainty can be accurately described by a known probability distribution (or class of probability distributions, such as gaussians), which may not always be realistic. Depending on the specific problem setting, one approach may offer clearer advantages over the other. This trade-off has been widely discussed in the literature (see for example Ben-Tal et al. 2009, Bertsimas et al. 2011, Delage and Ye 2010).
> >
> > - **Computational tractability.** Distribution-based uncertainty often presents significant computational challenges. Solving optimization problems under distributional assumptions typically requires evaluating (and optimizing) expectations, variances, or risk measures—quantities that often lack closed-form expressions and must be approximated through sampling or Monte Carlo methods. These techniques become particularly costly in high-dimensional settings. Perhaps the closest concept in Bayesian decision theory to our setting is Bayesian Experimental Design (BED), which also suffers from high computational cost and poor performance under model misspecification. As noted in (Rainforth et al. 2024), “perhaps the biggest challenge here, and indeed a key challenge for BED more generally, will be scaling to larger and more complex problems. Despite recent progress, current methods remain limited by factors such as the dimensionality and smoothness of the model and design space, as well as the experiment horizon T". In contrast, set-based uncertainty models often lead to more tractable formulations (see (Bertsimas et al. 2018), (Ben-Tal et al. 2002)).
> >
> >   To be more concrete, *one example of constructing uncertainty sets that uses both milder distributional assumptions and enjoys tractability* properties is the technique Bertsimas et al. (2018), "Data-Driven Robust Optimization". In that paper, the authors construct uncertainty sets by first specifying limited structural assumptions on the true distribution—such as independence across components, bounded support, or availability of marginal samples—and then applying statistical hypothesis tests to data drawn i.i.d. from this distribution. The uncertainty set is derived by taking the confidence region of the test, that is, the set of distributions that would not be rejected at a chosen significance level, and computing a worst-case bound over this region. This approach yields uncertainty sets that are computationally tractable and require milder assumptions than a fully specified distribution (or distribution class) of the uncertainty.

---

> > > ### Author Response · Authors · 2025-08-07
> > > **On the practicality of set-based uncertainty modeling (Part 2)**
> > >
> > > **Part 2/2**
> > >
> > > - **A vast number of successful practical applications.** The paper "A practical guide to robust optimization" from Gorissen et al. in 2015, cited over 800 times, details the practicality of such modeling and argues that it *"is very useful for practice, since it is tailored to the information at hand, and it leads to computationally tractable formulations"*. Several researchers and practitioners have indeed applied set-based uncertainty modeling, through robust optimization for example, in practical applications. Examples of these include (Paul and Wang 2019) for network design, (Balcik and Yanıkoğlu 2020) and (Ben-Tal et al. 2020) for logistics planning, (Stienen et al. 2021) for facility location, (Malcolm and Zenios 1994), (Nazari-Heris 2018), and (Verástegui et al. 2019) for power systems, to mention a few. A flurry of more applications can be found in several surveys and books on robust optimization, one of the more recent being "Robust and adaptive optimization" of Bertsimas and Den Hertog (2022).
> > >
> > > Thank you again for all the efforts in reviewing our paper. If the reviewer deems it useful, we can include a similar discussion of our modeling choice in a revised version of the paper.
> > >
> > > ### References
> > > 1- (Ben-Tal et al. 2009) Ben-Tal, El Ghaoui, and Nemirovski (2009) — Robust Optimization
> > >
> > > 2- (Bertsimas et al. 2011) Bertsimas, D., Brown, D. B., & Caramanis, C. (2011). Theory and applications of robust optimization. SIAM review, 53(3), 464-501.
> > >
> > > 3- (Delage and Ye 2010) Delage, E., & Ye, Y. (2010). Distributionally robust optimization under moment uncertainty with application to data-driven problems. Operations research, 58(3), 595-612.
> > >
> > > 4- (Bertsimas et al. 2018) Bertsimas, Dimitris, Vishal Gupta, and Nathan Kallus. "Data-driven robust optimization." Mathematical Programming 167.2 (2018): 235-292.
> > >
> > > 5- (Ben-Tal et al. 2002) Ben-Tal, Aharon, and Arkadi Nemirovski. "Robust optimization–methodology and applications." Mathematical programming 92.3 (2002): 453-480.
> > >
> > > 6- (Rainforth et al. 2024) Rainforth, Tom, et al. "Modern Bayesian experimental design." Statistical Science 39.1 (2024): 100-114.
> > >
> > > 7- (Paul and Wang 2019) Paul, J. A., & Wang, X. J. (2019). Robust location-allocation network design for earthquake preparedness. _Transportation research part B: methodological_, _119_, 139-155.
> > >
> > > 8- (Balcik and Yanıkoğlu 2020) Balcik, B., & Yanıkoğlu, İ. (2020). A robust optimization approach for humanitarian needs assessment planning under travel time uncertainty. _European Journal of Operational Research_, _282_(1), 40-57.
> > >
> > > 9- (Ben-Tal et al. 2011) Ben-Tal, A., Do Chung, B., Mandala, S. R., & Yao, T. (2011). Robust optimization for emergency logistics planning: Risk mitigation in humanitarian relief supply chains. _Transportation research part B: methodological_, _45_(8), 1177-1189.
> > >
> > > 10- (Stienen et al. 2021) Stienen, V. F., Wagenaar, J. C., Den Hertog, D., & Fleuren, H. A. (2021). Optimal depot locations for humanitarian logistics service providers using robust optimization. _Omega_, _104_, 102494.
> > >
> > > 11- (Gorissen et al. 2015) Gorissen, B. L., Yanıkoğlu, İ., & Den Hertog, D. (2015). A practical guide to robust optimization. _Omega_, _53_, 124-137.
> > >
> > > 12- (Nazari-Heris 2018) Nazari-Heris, M., & Mohammadi-Ivatloo, B. (2018). Application of robust optimization method to power system problems. _Classical and recent aspects of power system optimization_, 19-32.
> > >
> > > 13- (Malcolm and Zenios 1994) Malcolm, S. A., & Zenios, S. A. (1994). Robust optimization for power systems capacity expansion under uncertainty. _Journal of the operational research society_, _45_(9), 1040-1049.
> > >
> > > 14- (Verástegui et al. 2019) Verástegui, F., Lorca, A., Olivares, D. E., Negrete-Pincetic, M., & Gazmuri, P. (2019). An adaptive robust optimization model for power systems planning with operational uncertainty. _IEEE Transactions on Power Systems_, _34_(6), 4606-4616.
> > >
> > > 15- (Bertsimas and Den Hertog 2022) Bertsimas, D., & Den Hertog, D. (2022). Robust and adaptive optimization, *Dynamic Ideas*

---

### Note · Authors · 2025-08-12

We are grateful for the AC and all the reviewers' time spent on our paper. We would like to thank reviewer zZVtm, jqbm and At1w for their scores recommending acceptance and for acknowledging our rebuttal.

We hope our last response to reviewer zZVtm addresses their concern about set-based modeling of uncertainty: an established approach in the robust optimization literature (with several books and a rich literature) rather than a novelty of our paper.

It is unfortunate that reviewer 6K14 did not change their rejection score after a lengthy discussion that established several misunderstandings and incorrect claims about our paper in their review. Below is a brief response to their last message outlining reasons for rejection, which came shortly before the deadline.

- **Motivation:** We believe the question of data informativeness to be of major importance as we motivate in the introduction. Understanding which data enables optimal decisions is "an important question of ML" and "a fundamental and important question at the core of data-driven decision-making" as Reviewer jqbm and At1w state, respectively. It allows for instance to understand which data to collect when building a new model for a given task.
- **Literature Review:** As detailed in the rebuttal, the prediction-and-optimize literature addresses a fundamentally different question, both in motivation and technical treatment. The authors would be happy to include a discussion in the revision. We do not believe its absence to be "a significant weakness" justifying rejection. It is connected to our work through the study of LPs (as is inverse optimization, LP queries, online LO...), but in a 9-page submission, we decided to focus on literature studying the data question, our core focus and novelty.
- **On Algorithm 1:** It is incorrect that Algorithm 1 "solves an LP optimization for every possible parameter vector". The algorithm is described in detail in Section 4. It is immediate from its output (a basis of a subspace of $\mathbb{R}^d$, one vector per iteration), that it can only run for at most $d$ iterations (a basis is always of size $\leq d$). Alg 1 is indeed the one being solved. Alg 2 just details how to perform each step of Alg 1.
- **On Experiments:** We believe to be the first to address such a problem. Therefore, there are no baseline competitors to compare with. The value of the paper is in the theory developed and the experiments are only meant to illustrate the theory in an example.

---

### Decision · Program_Chairs · 2025-09-17

**Decision:**

Accept (poster)

**Comment:**

This paper studies linear programs where problem parameters are unknown and lie in an uncertainty set. Given a dataset that provides information about these parameters, the paper characterizes when this dataset is sufficient to identity an optimal solution.

The strength of this paper is that it is a clearly-written theory paper studying a novel question of data informativeness in the context of an important problem class (linear programs).

Much of the discussion centered around aspects of linear programming and robust optimization that stem from the novelty of these techniques within the NeurIPS community.  The authors handled these points well in their rebuttal. Let me list some of them:
- Questions about the novelty of the analysis relative to work on sensitivity analysis.  As the authors explain, both this paper and works on sensitivity analysis use similar technical tools, but this doesn't significantly detract from the paper's novelty.
- Concerns about how to choose the uncertainty set. As the authors explain in their rebuttal, this is a well-studied question in the robust optimization literature and the authors should not be faulted for this concern.
- The work does not study nonlinear programs. I do not view this as a problem. There are many important problems that can be formulated as linear programs.

The reviews also raised points that I view as real weaknesses. I list them here:
- Computational cost of solving many MILPs. The authors point out that there are many problems where the cost would be acceptible, and that the number of MILP variables and constraints have a reasonably benign scaling. Thus, while this is a limitation, there are still many problems where the proposed method could be useful.
- The paper's results focus on the noiseless setting and become more limited when there is noise.

Reviewer 6K14 rated the paper "2 reject" but the reasons cited seem to stem from a difficulty understanding the paper and not from a weakness in the paper itself. While there are always opportunities to improve clarity in a paper, the writing in this paper is reasonably clear. Instead, the low score seems to stem from the fact that linear programming is not well known to many reviwers at NeurIPS. I view bringing ideas from linear programming to NeurIPS as a strength of this paper, not a weakness, and so I am discounting 6K14's rating.